# Statistical and Radio Analysis of Exoplanets and Their Host Stars

Baoda Li [1,2], Liyun Zhang [1,2,*], Tianhao Su [1,2], Xianming L. Han [3], Prabhakar Misra [4] and Liu Long [5]

1. College of Physics, Guizhou University, Guiyang 550025, China; gs.bdli21@gzu.edu.cn (B.L.); su.tian.hao@outlook.com (T.S.)
2. Guizhou Provincial Key Laboratory of Public Big Data, Guiyang 550025, China
3. Department of Physics and Astronomy and SARA, Butler University, Indianapolis, IN 46208, USA; xhan@butler.edu
4. Department of Physics and Astronomy, Howard University, Washington, DC 20059, USA; pmisra@howard.edu
5. Department of Astronomy, Beijing Normal University, Beijing 100875, China; longliu@mail.bnu.edu.cn
* Correspondence: liy_zhang@hotmail.com

**Abstract:** As of February 2022, over 4900 exoplanets have been confirmed. In this study, we conducted statistical analyses on both the exoplanets and their host stars' parameters. Our findings suggest that the radius and true mass distribution of the exoplanets remain largely unchanged compared to prior research. However, we observed a correlation between the average eccentricity and the number of planets in a system, and fluctuations in the "size" of the planets may contribute to such variation. Moreover, we discovered that, among planets with precise measurements of radius, true mass, and semi-major axis, the true mass-radius relationship follows a power–law distribution. Interestingly, the power–law index tends to decrease from super-Earths to cold Jupiters, potentially due to atmospheric composition. We also revised the radius valley, and determined that M-type host stars with low mass and metal abundance exhibit high planetary ownership rates or harbor large-mass planets, suggesting a different planet formation mechanism than GK-type stars. Lastly, we assessed the possibility of detecting exoplanets using FAST and found that there are three planets in FAST sky that may be detected, namely CoRoT-3 b, GPX-1 b, and TOI-2109 b.

**Keywords:** exoplanets: exoplanets; stars: planetary systems; stars: statistics

## 1. Introduction

In 1995, ref. [1] discovered the first exoplanet orbiting a sun-like star (non-neutron) in the Pegasus constellation, named 51 Pegasi b, which marked the beginning of exoplanet science. As the number of planets discovered continues to increase, researchers are constantly exploring the characteristics of planets and their host stars. However, as planet radii and semi-major axis of planetary orbits increase, the number of planets discovered decreases [2,3]. Ref. [4] found that the number of known exoplanets decreases with an increase in planetary mass, which is consistent with the findings of [5], who studied the mass distribution of planets discovered by the radial velocity method.

Stars with known planets generally have higher metallicity than those without, indicating that metal-rich systems may be more favorable for planet formation [3,5–7]. Metallicity is believed to be an important factor in the evolution of galaxies and the history of star formation [8]. Spectral types of stars with exoplanets are typically F, G, K, and M stars [9]. Planetary parameters, such as eccentricity, semi-major axis, and mass, are influenced by planetary migration, interactions between planets, and surface temperature [5,10–13]. Planets in highly eccentric orbits may experience drastic seasonal changes that limit their habitability due to a lack of liquid water [13]. The radius of a planet is a key determinant of its size, and there is a power–law relationship between the mass and radius of

planets [14]. The radius valley, which refers to the scarcity of planets with radii between 1.6 and 2.2 $R_\oplus$ and periods less than 30 days, is an important feature in the radius–period relationship [15–18].

Studies have also shown that stars can affect the orbital inclination and eccentricity of planets through mutual Kozai–Lidov resonance, and the mass of the star can affect the formation and migration of planetary systems [4,12,19,20]. The semi-major axes of a star's planets generally increase with the star's mass [21], and the number of low-mass planets tends to decrease with the increasing star mass [22]. Ice giants close to Neptune's mass may be more common around small stars, while gas giants close to Jupiter's mass are relatively rare [23].

Radio emission can provide valuable insights into the magnetic fields of planets and their host stars, allowing for the measurement of physical parameters and the detection of exoplanets. A large-scale comparative study on the predicted radio emission characteristics of alien clusters in known departments can help to determine the physical parameters responsible for producing bright and observable radio emissions. The planetary radio emission can propagate through the stellar wind, which requires a planetary magnetic field strength greater than 1.3–13 G [24,25]. Moreover, detecting radio emissions from exoplanets offers a new method for directly detecting exoplanets and measuring their magnetic field intensity and rotation period [26]. In 2004, Joseph et al. developed a model to evaluate the radio flux density generated by the interaction between planets and their host stars, which revealed that radio emission was mainly located in the low-frequency band, presenting a significant challenge for radio detection equipment [27].

In recent years, the study of exoplanets has been rapidly expanding, and as of February 2022, more than 4900 exoplanets have been discovered through various methods. The official website (https://exoplanetarchive.ipac.caltech.edu/ (accessed on 15 February 2020)) provides basic information on exoplanets and their host stars, presenting an ample dataset for the statistical analysis of the characteristics of exoplanets and their host stars. In this paper, we used the data containing duplicates from 32,112 of 4933 exoplanets. To ensure the accuracy of our analysis, we selected samples that meet specific criteria such as relative error range and parameters available. While some parameters such as mass and radius may not be available for all exoplanets due to observational limitations, the larger sample size allows for a more complete parameter space to be explored, providing further insight into the properties of exoplanets. To ensure the originality of the data used in this study, we did not include those data points without both mass and radius measurements even though the predictor of [28] can provide some estimates.

In this study, we conducted a comprehensive analysis of exoplanetary systems by exploring the distribution of planetary parameters in Section 2, and calculated the radio emission properties of the exoplanetary systems in Section 3, Finally, our discussion and conclusions are presented in Section 4.

## 2. Parameter Distribution Statistics of Planets and Host Stars

### 2.1. Selection of Samples

To ensure the accuracy and reliability of our statistical analysis, we carefully select parameters with low relative errors as real samples. The relative error is calculated using the formula:

$$relative\ error = \frac{data\ error}{data\ value} \tag{1}$$

We consider data with a relative error of less than 10% as reliable. However, for a broader range of statistical samples, we may select samples with a relative error range of up to 50%, based on the total sample size. While we did not use this criterion for the statistics of a single parameter in the second section, we used the total parameter sample and added the relative error bar. After the second part, we filter the data using this standard to ensure the quality of our analysis.

## 2.2. Host Star System with Extrasolar Planets

The formation of a star system occurs when stars attract other materials to surround them through their own gravity or when multiple stars are surrounded by each other and attract surrounding materials [29]. By analyzing data related to discovered planets, we obtained statistical results for star systems, as shown in Table 1. The highest number of exoplanets has been found in single-star systems, with the number of discovered planets decreasing as the number of stars in the system increases. In previous surveys, the system with a simpler structure was often chosen to reduce complexity, leading to a higher incidence of planets in single-star systems. Within different star systems, single-planet systems account for the largest proportion. This is partly due to the fact that the complexity of a system may decrease the probability of planet discovery, and some planets remain undiscovered. Additionally, the surrounding material environment plays a role in the generation of planets. It is also important to note that the number of stars in a system can affect the dynamics of planet formation [19].

**Table 1.** Distribution of planetary system multiplicity.

|             | **Single Planet** | **Double Planets** | **Multi-Planet** |
| --- | --- | --- | --- |
| Single star | 2618   | 496    | 248   |
| Double star | 233    | 36     | 30    |
| Multi-star  | 36     | 5      | 2     |
| Percentage  | 77.94% | 14.50% | 7.56% |
| Total       | 3704   |        |       |

## 2.3. Planetary Related Parameters

In the field of planetary science, the parameters related to planets are crucial for understanding the properties and characteristics of planetary systems. These parameters mainly include the mass, radius, semi-major axis, eccentricity, and metal abundance of planets. In recent years, advances in computational science and databases have enabled the more accurate determination of these parameters [30]. Detailed analysis and understanding of these parameters can provide valuable insights into the formation and evolution of planetary systems, as well as their current status.

### 2.3.1. Distribution of True Mass and Radius of Planets

We only selected the planets with true mass and radius measurements from the total samples, and we chose the latest available data for each planet sample. To ensure accuracy, we only included samples with relative errors less than 0.5 based on Formula (1) due to the large uncertainty of many true mass measurements. At the same time, because the sensitivity of ground detection and space detection to different planets is different, we distinguish space samples and ground samples and analyze the true mass (Figure 1) and radius distribution (Figure 2) of planets according to different planetary search methods, and draw the logarithmic scale. Due to the influence of the Earth's atmospheric disturbance, the sensitivity of ground facilities to the population of small-mass and small-radius planets is seriously reduced, so in the distribution of ground samples, small-mass and small-radius planets are rarely seen. Ground facilities are more sensitive to planets with a mass of about 300 $M_\oplus$ and a radius of about 15 $R_\oplus$ (about Jupiter). On the contrary, due to the vacuum in space, space facilities have a better line of sight, so they are more sensitive to small-mass and small-radius planets (the peak of mass distribution is about 9 $M_\oplus$. The mass samples in space indicate the shortage of planets near $log_{10}(M_P)$ ranging from 1.1 to 2.1, with a medium mass range of 10–100 $M_\oplus$ and a radius of 6 $R_\oplus$) (the logarithmic value is 0.8), which is the statistical characteristic of the sub-Saturn desert. Due to the limited sensitivity of ground facilities, we cannot find it in ground samples. In the radius distribution of

space samples, planets are concentrated around the radius of 1.5 $R_\oplus$ (logarithmic value of 0.2) and 3 $R_\oplus$ (logarithmic value of 0.5), which is the typical radius of super-Earth and sub-Neptune. A small valley between super-Earths and sub-Neptunes separates the two populations and will become more evident with the increase in samples. The valley may indicate different formation and evolution mechanisms for sub-Neptunes and super-Earths. Small planets are commonly formed in protoplanetary disks, but only a small fraction of them grow to a critical size over time and become gas giants. Moreover, low-mass planets at close distances may experience significant light evaporation and lose a considerable portion of their original H/He inventory, resulting in a smaller radius [31–33]. This causes planets to accumulate at small and medium radii. [34] predicted that 23% of stars have a planet with a mass similar to Earth's (0.5–2.0 $M_\oplus$). Among different search methods, the transit method is undoubtedly the most successful, because it is not limited by facilities and finds most of the planets in the total sample, especially in the space survey, while other search methods are mainly in the ground survey, such as the RV method. The imaging method is very sensitive to massive planets in wide orbits. Therefore, most of the planets detected by this technology have large radii and high masses.

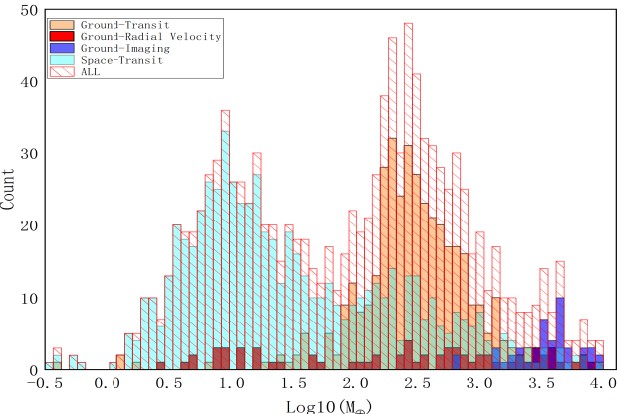

**Figure 1.** Histogram of true mass distribution of known exoplanets. The abscissa is the logarithm of radius, and the ordinate is the number of planets in each box with a logarithm of the radius. Among them, different color bars are used to distinguish the samples of space, ground, and different discovery methods.

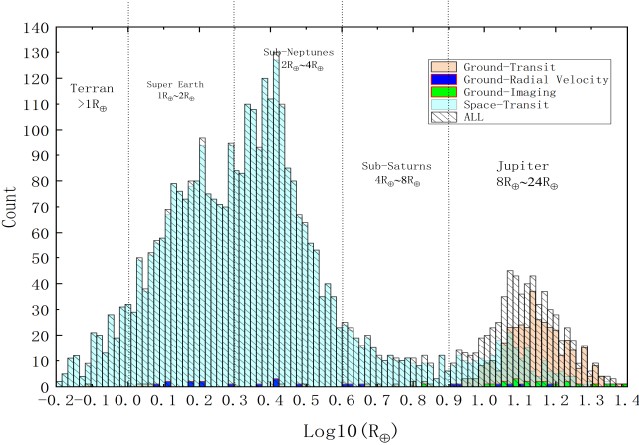

**Figure 2.** Histogram of the radius distribution of known exoplanets. The abscissa is the logarithm of mass of planets, and the ordinate is the number of planets in each box with a logarithm of mass. Among them, different color bars are used to distinguish the samples of space, ground, and different discovery methods. Planets are classified by a number of black dotted lines.

### 2.3.2. Number of Planets and Eccentricities in the System

Planetary systems exhibit varying eccentricities, which can have significant effects on their dynamics, internal structure, and climate. Highly eccentric orbits result in dramatic seasonal variations in surface temperatures due to varying stellar radiation, potentially rendering them uninhabitable by restricting the presence of liquid water on their surfaces [13]. As a result, low-eccentricity planets are more likely to be habitable than their highly eccentric counterparts. The planetary orbits are primarily determined by the interaction between planets, with different planets in a system exerting gravitational forces on each other. Such interactions can disturb the orbits of the planets and cause variations in the eccentricities of the entire system, especially when there is a significant size difference between the planets [11].

Measuring the eccentricity of individual Kepler planets has been mainly performed through the modeling of the TTV and TDV signals. However, such models may introduce a bias in the sample of selected planets, limiting the representativeness of the derived eccentricity distribution. To address this issue and explore the stability of eccentricity in multi-planet systems, we searched for systems with a radius, true mass, and eccentricity measurements among the total sample of planets. We found 42 systems with two planets, 23 with three planets, 9 with four planets, 3 with five planets, and 3 systems (Kepler-11, K2-138, and TRAPPIST-1) with six and seven planets, respectively (without any restrictions on the host star). Larger planets need more gravity to restrain them. If there is a planet with a larger scale in the system, will it have a greater impact on the orbit of the whole system? To quantify the radius and eccentricity fluctuations in each system, we used the logarithm $(Rp_{Max}) -$ logarithm $(Rp_{Average})$ and logarithm $(Ep_{Max}) -$ logarithm $(Ep_{Average})$ values, respectively. $Rp_{Average}$ is the average radius of all planets in the same system, while $Ep_{Average}$ is the average eccentricity of all planets in the same system, and $Rp_{Max}$ and $Ep_{Max}$, respectively, represent the maximum radius and eccentricity of planets in the system. As shown in Figure 3, the larger fluctuation of planet "size" is often related to the higher eccentric fluctuation in the system, and the system with fewer planets tends to show smaller eccentric fluctuation on average. The uneven distribution of planet "size" may impact the stability of planetary orbits. Although the selected samples are few for the whole sample, which may cause large errors, we consider that this may be a direction that can be of concern, that is, whether the eccentricity of the planets in the multi-planetary system may be disturbed by the larger-scale planets, which requires more complete data samples to discuss this.

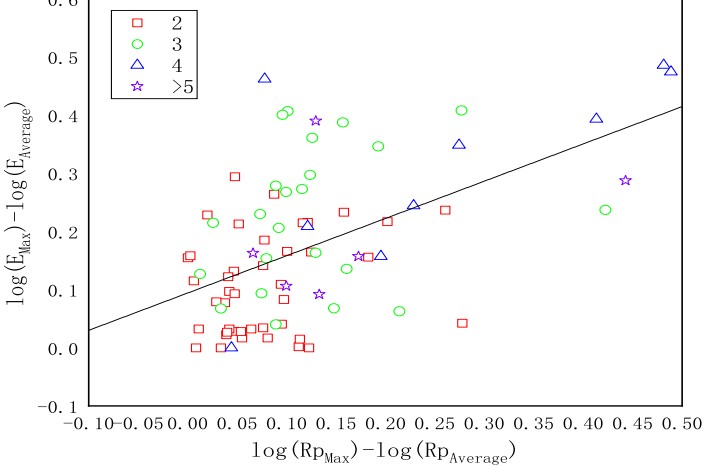

**Figure 3.** Radius fluctuation vs. eccentric fluctuation of planets. The abscissa is explained as the fluctuation of the system planet radius, and the ordinate is explained as the fluctuation of the system planet eccentricity. Red square (2 planets), green circle (3 planets), blue triangle (4 planets), purple stars (5+ planets). The trend line is shown in black.

We further studied the average eccentricity of different planetary systems, as shown in Figure 4, where the average eccentricity is the average eccentricity of all samples in all planetary systems with the same number of planets. In systems with fewer planets, the interactions between planets will decrease, and the decrease in constraints will increase the complexity of orbital evolution, leading to a need for higher eccentricities to maintain crossed orbits. Conversely, in systems with more planets, the last planet needs to be formed with an orbit far away from the intersection of other planets, leading to smaller eccentricity dispersions [35]. This finding is also supported by a recent study using the transit duration method [18] and is qualitatively consistent with the results obtained from the RV planet. In our samples (RV samples are less than 10% of the total sample), the average eccentricity also tends to decrease with the increase in the number of planets in the system, which further confirms this point.

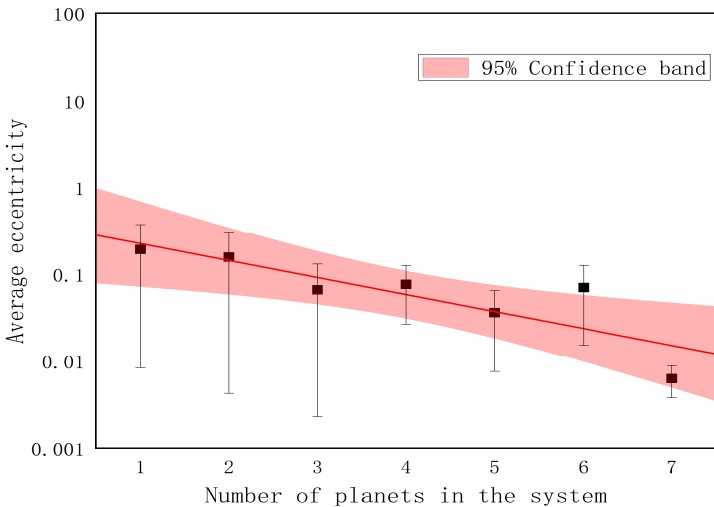

**Figure 4.** Correlation between the number of planets in a system and the average eccentricity of planets. The abscissa is the average eccentricity of the planetary system, and the average eccentricity of the system is carried out in all planetary samples of systems with the same planet. The ordinate is the number of planets in the system. The trend line is shown in red. The red area is the 95% confidence band of the trend line, and the standard deviation of the data is displayed on the error bar.

2.3.3. Planet Mass and Planet Radius

Ref. [36] investigated the relationship between the mass and radius of planets using average density as a metric. Their analysis indicated that there may be a correlation between the two parameters based on direct fitting. Similarly, Ref. [14] observed that the mass–radius relationship of planets can be described by a piecewise power–law exponent to some degree.

$$\frac{R}{R_{\oplus}} = C(\frac{M}{M_{\oplus}})^{\beta} \tag{2}$$

Here, we have carefully selected a sample with a relative error of less than 10% between the radius, true mass, and semi-major axis of the orbit. We refer to the mass–radius relationship for rocky planets by [37] for our analysis, which is defined for planetary core mass fractions (*CMFs*) ranging from 0 to 0.4 and planetary masses ($1M_{\oplus} \leq M \leq 10M_{\oplus}$). The relationship is mathematically expressed as:

$$(\frac{R}{R_{\oplus}}) = (1.07 - 0.21CMF)(\frac{M}{M_{\oplus}})^{1/3.7} \tag{3}$$

In addition to the mass–radius relationship related to *CMF* by [37], we also included the analytical relationship function between the mass and radius of a planet and the contents of water ice and rock in the planet, as presented by [38]:

$$\frac{R}{R_\oplus} = (0.0912 imf + 0.1603)(lg\frac{M}{M_\oplus})^2 + (0.3330 imf + 0.7387)(lg\frac{M}{M_\oplus}) + (0.4636 imf + 1.1193) \tag{4}$$

$$\frac{R}{R_\oplus} = (0.0592 rmf + 0.0975)(lg\frac{M}{M_\oplus})^2 + (0.2337 rmf + 0.4938)(lg\frac{M}{M_\oplus}) + (0.3102 rmf + 0.7932) \tag{5}$$

Here, *imf* and *rmf* denote the mass ratios of water ice and rock in the planetary structure, respectively. A value of 1 for *imf* corresponds to a planet made entirely of water ice, while values of 0 or 1 for *rmf* indicate a planet made entirely of rock or iron, respectively.

In Figure 5, the samples in our study with relatively small errors follow diagonal bands from low mass/small size to high mass/large size. Around 210 $M_\oplus$, they reach a smooth maximum and begin to float on the corresponding radius. Small planets are commonly formed in most protoplanetary disks, where protoplanets must grow to masses of 5–10 $M_\oplus$ by the accretion of solids (dust and ice) from the disk. The protoplanet then undergoes runaway gas accretion, increasing its mass by an order of magnitude, but only if the protoplanetary gas has not yet dissipated [39]. A small portion of these protoplanets grows to a critical size over time and becomes gas giants. However, in situ formation is unlikely for hot Jupiters due to the insufficient protoplanetary disk mass so close to the star. Instead, they likely formed in the disk at several AU, were gravitationally perturbed into orbits with random inclinations and high eccentricities, and were captured at 0.05 AU by the dissipation of orbital energy in tides raised on the planet [39].

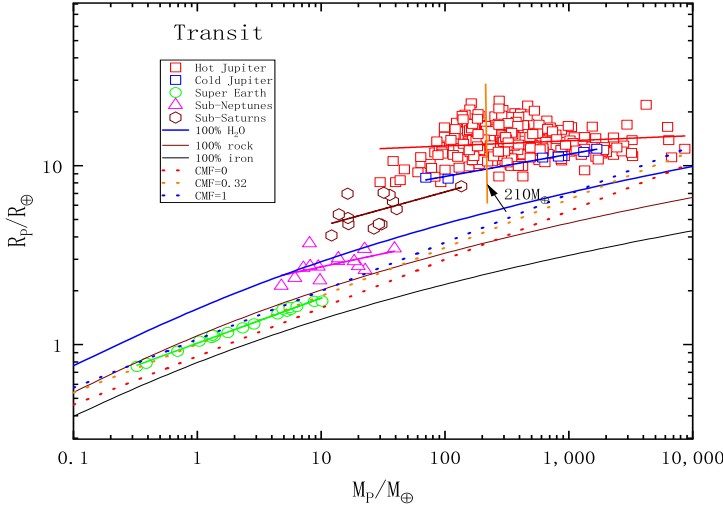

**Figure 5.** Mass–radius relationship of exoplanets. The red square is hot Jupiter, the blue square is cold Jupiter, the green circle is super-Earth, the purple triangle is sub-Neptune, and the brown hexagon is sub-sub-Saturn. The dotted line shows the mass–radius relationship of the rocky planet by [37]. The solid line is [38].

Table 2 shows the average orbital semi-major axis of each population in the selected sample. The average orbital semi-major axis of hot Jupiters in our sample is around 0.56 AU, which is consistent with previous studies. The fitting parameters of the M-R relationship of different populations are also shown in Table 2. In the fitting of super-Earths, we obtain a fitting relation index of approximately 0.25, which is consistent with that obtained by [37] (1/3.7). When *CMF* = 0.3, the M–R relationship can well pass through our super-Earth sample, indicating that the core mass fraction is very close to *CMF* $\simeq$ 2.6, which suggests that the ratio of their mantle and core is similar to that of Earth. However, their surface

conditions are completely different because they are too hot. This is due to observational bias, as it is easier for astronomers to detect planets that are closer to their host stars. However, based on the evidence we gathered, it appears that there may be a significant number of Earth-like planets located at an appropriate distance from their stars, allowing for the presence of liquid water on their surfaces. The selected samples in our study, except for cold Jupiter planets, have an average orbital semi-major axis of less than 0.1 AU. These planets are very close to their host stars and are strongly radiated, making them more likely to be naked rocky planets or have a thin atmosphere. According to the analysis curve of [38], the super-Earths in the sample are within the range of rocky planets. If a planet with a larger size has a larger semi-major axis of the orbit, the atmosphere of the evaporating planet radiated by the illumination of the host star will gradually decrease, and the atmosphere of the planet will occupy a large part of the radius. When the mass of the planet is 210 $M_\oplus$, the size of the planet does not increase with the increase in H/He atmospheric mass, but with the increase in H/He atmospheric mass fraction. A 100 $M_\oplus$ planet with a 10 $M_\oplus$ core is already 95% H/He, so the increase in planet mass will not significantly increase the radius of the planet [33]. It can be observed from Table 1 that the $\beta$ of the population relationship fitting gradually decreases with an increase in planet mass. Furthermore, with an increase in planetary size, there appears to be a decrease in the correlation between mass and radius in the mass–radius relationship, which could potentially be attributed to the degree of influence that the planet's core and atmosphere have on changes in its radius.

**Table 2.** Fitted parameters for the relationship between planetary mass and radius.

| Type | Average Semi-Major Axis (Au) | $X(C = 10^X)$ | $\beta$ | $R^2$ | Number of Planets |
|---|---|---|---|---|---|
| Super Earth ($0.6R_\oplus$–$2R_\oplus$) | $0.03422889 \pm 0.000668125$ | $0.00769 \pm 0.00277$ | $0.2522 \pm 0.0048$ | 0.99389 | 20 |
| Sub-Neptunes ($2R_\oplus$–$4R_\oplus$) | $0.07134143 \pm 0.0014143$ | $0.28608 \pm 0.07047$ | $0.14827 \pm 0.06305$ | 0.31543 | 14 |
| Sub-Saturns ($4R_\oplus$–$8R_\oplus$) | $0.0877815 \pm 0.00164923$ | $0.47008 \pm 0.11561$ | $0.19154 \pm 0.07928$ | 0.34666 | 13 |
| Cold Jupiter ($8R_\oplus$24$R_\oplus$, >0.1AU) | $0.97186 \pm 0.0127614$ | $0.69102 \pm 0.02045$ | $0.12424 \pm 0.00746$ | 0.98228 | 7 |
| Hot Jupiter ($8R_\oplus$24$R_\oplus$, <0.1AU) | $0.0563939 \pm 0.00111049$ | $1.05251 \pm 0.03032$ | $0.02877 \pm 0.01169$ | 0.01978 | 302 |

2.3.4. Planetary Radius and Period (Radius Valley)

The radius valley phenomenon can be explained by the theory of light evaporation, where the high-energy photons of the host star cause the planet's atmosphere to evaporate, separating the hydrogen and helium atmosphere from the planet's surface and reducing its radius. The valley separates planets with and without an expanded atmosphere [32]. To accurately locate the valley, we analyzed planetary samples discovered by transit and applied strict criteria, including limiting the relative errors of radius, period, and mass of host stars to 10%, excluding stars (M < 0.45 $M_\oplus$), because the formation mechanism of planets around M dwarf stars may be different from that around FGK stars. Meanwhile, in order to ensure that only one host star in each system can affect the planets around it, only one host star will be allowed in each system. We obtained 1213 samples and did not include relative error bars in the statistical charts due to their small size.

The radius measurements shown in the figures are calibrated by [18],

$$\frac{R_p}{R_p^{valley}} = \left(\frac{P}{10days}\right)^g \left(\frac{M_\star}{M_\odot}\right)^h \tag{6}$$

$$R = R_P \left(\frac{P}{10days}\right)^{-g} \left(\frac{M_\star}{M_\odot}\right)^{-h} \tag{7}$$

Here, $Rp$ denotes the radius of the planet, $p$ represents the period of the planet, $M_\star$ is the mass of the host star, $g$ is the quantitative period dependence index, and $h$ is the quantitative star mass dependence index. According to previous reports, the radius position of the valley decreases with the increase in the orbital period [17] and increases with the

increase in the host star mass [40]. When the orbital period $P = 10$ days and the host mass $M = 1\,M_\odot$, the valley was found in $R_p^{valley} = 1.9 \pm 0.2 R_\oplus$. The slope quantifying the period dependence is $g = -0.09^{+0.02}_{-0.04}$ [17]. The slope quantifying the stellar mass dependence is $h = 0.26^{+0.21}_{-0.16}$ [40]. Through the above relationship, we can recalculate the radius. Figure 6 shows the relationship between the planetary period and the calibrated radius. Moreover, Figure 7 shows the relationship of the incidence of planets (number of planets in regional samples divided by the total number of host stars) and the calibrated radius.

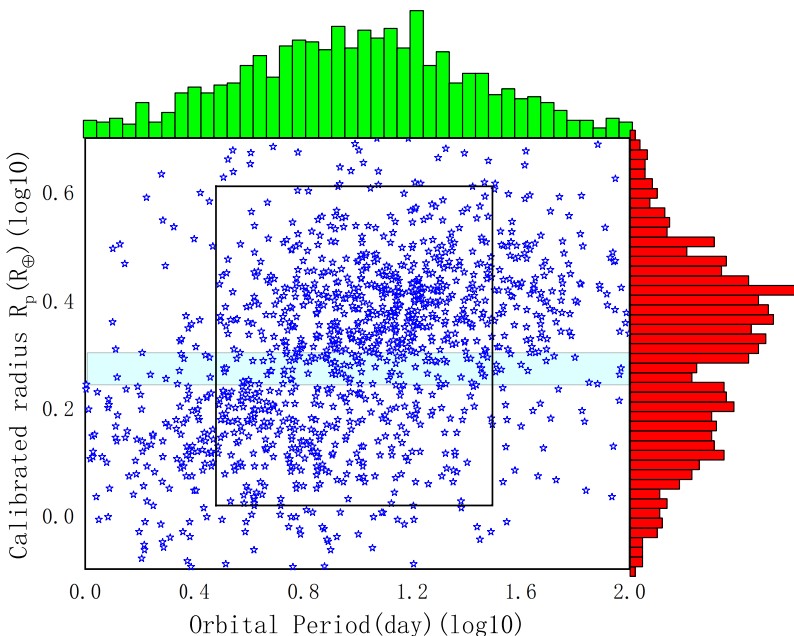

**Figure 6.** Relationship between planetary period and calibrated radius. The abscissa is the logarithm of the orbital period and the ordinate is the calibration radius. The upper diagram shows the orbital period distribution of the sample, and the right diagram shows the calibration radius distribution. The light blue bag in the middle marks the sparse area of the planet, which corresponds to the obvious valley in the calibration radius distribution on the right. The black box in the middle is a sample with a period of 3–30 days and a calibration radius of 1–4 $R_\oplus$, which will be used for statistics and drawing Figure 7.

Figure 6 illustrates a small gap (light blue region) near 2 $R_\oplus$ during the period of 3–30 days. Figure 7 is derived from the data in the black box of Figure 6 and highlights the radius valley (blue range). It is basically consistent with previous reports [18,41], and the radius gap is more obvious when the dependence of the period and host star mass are considered simultaneously. The region of the valley here is about 1.7–1.9 $R_\oplus$ (blue vertical band). The valley between two peaks with uncalibrated radius is narrower than that with calibrated radius, and the peaks on both sides of the valley with a calibrated radius are 1.5 $R_\oplus$ and 2.6 $R_\oplus$, respectively. Compared with [18], the peak at $R < R_p^{valley}$ is more obvious, which might be caused by the increase in samples.

The radius valley can be generated by two main mechanisms: one is the previously mentioned light evaporation of the atmosphere, and the other is the core-driven mass loss mechanism proposed by [41,42]. In this mechanism, the driving factors for atmospheric separation are primarily derived from the internal luminosity of the cooling core, but it takes more time. Compared with the core-driven mass loss model, the valley obtained by the calibration radius here is narrower, but the shapes and positions of the two peaks are close. In the explanation of the core-driven mass loss mechanism by the core, the optical envelope of the planet is optically thinner, so it cools and shrinks quickly, while the heavy envelope cools slowly, roughly maintaining its initial radius. By eroding the

light atmosphere, the mass loss caused by core cooling amplifies this effect and will deepen the valley [32], as predicted by a narrow valley in the light evaporation model, and the threshold will be around 1.8 $R_\oplus$, which is in good agreement with our results. Such a narrow occurrence valley suggests that there may not be a large number of 1–10 $M_\oplus$ rocky planets formed by a huge crash after the disk has dissipated [43], and it is more likely to be migrated by light evaporation or core-driven mass loss mechanism. The character (for example, location and shape) of the radius valley mainly depends on the distribution of the planet's core mass, core composition, and atmospheric mass fraction [32]. If the location of the radius valley can be accurately determined, it will be helpful to determine the initial mass fraction critical value of the short-range (3–30 days) planet whose core mass is about the mass of the Earth (when high-energy radiation is enough to dissolve the whole H/He atmosphere) [44].

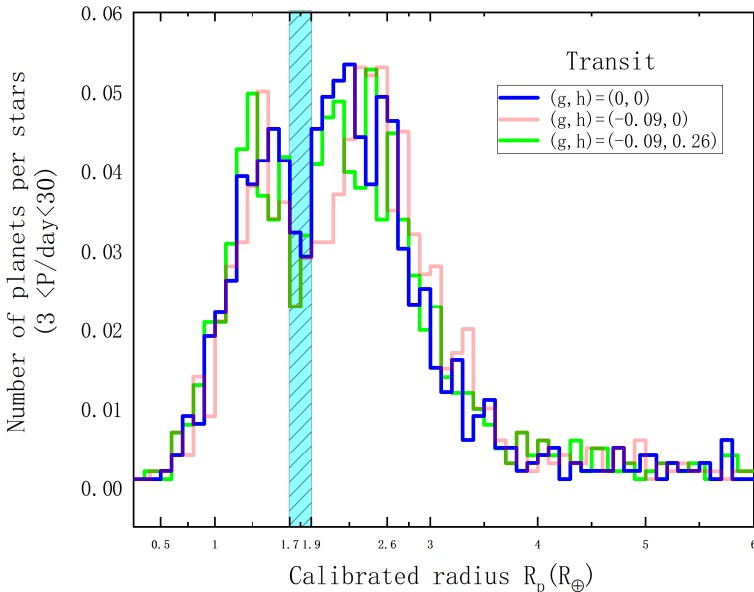

**Figure 7.** Relationship between the number of planets and calibrated radius. Here, the abscissa is the calibration radius. The histogram box here represents the ratio of the number of planets in each 0.1 step radius ranging from 0 to 30 $R_\oplus$ to the total number of stars in the sample system, and we added the explanation of the histogram box. In order to better show the radius valley, we only show the results of 0–6 $R_\oplus$. Different colors represent the calibration radius obtained using different parameters, and the blue vertical band marks the location of the radius valley.

### 2.4. The Mass of the Host Star and Its Planet

Previous studies have shown that the mass of the host star is a key factor in the formation and migration of planetary systems [4,12,20]. In order to further explore the relationship between the mass of the host star and the mass of its planets, we present Figure 8, which shows the mass measurements of exoplanets and their host stars. To ensure the reliability of the data, we limit the sample to systems with only one host star, and we also limit the relative error between the mass of the host star and the mass of the planet to 10%. The horizontal axis of Figure 8 represents the logarithm of the mass of the host star, while the vertical axis represents the logarithm of the planet's mass.

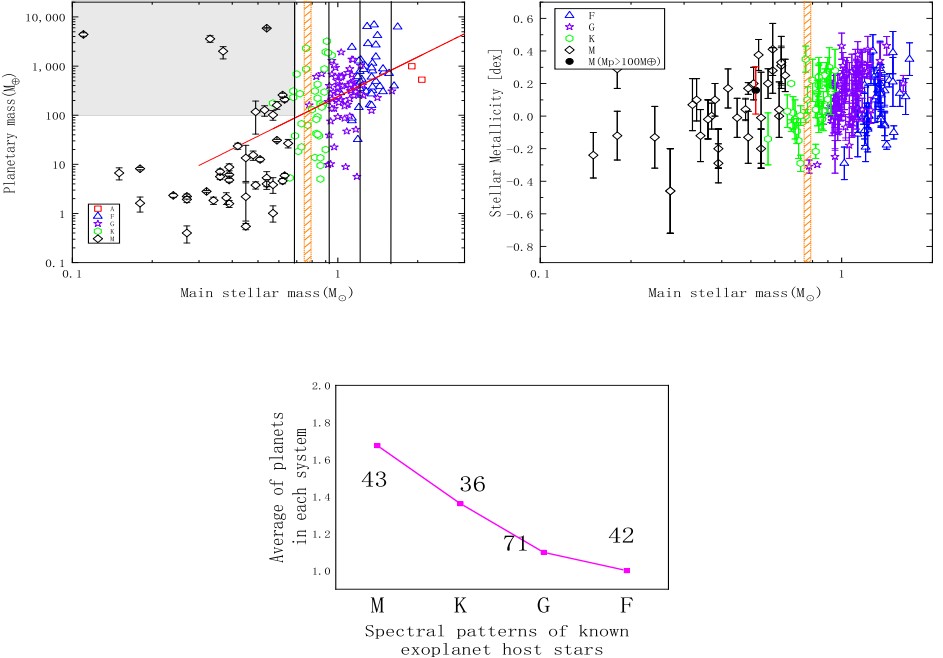

**Figure 8.** The left panel of Figure 8 displays the relationship between the mass of planets and the mass of host stars, the right panel displays the metallicity of these host stars, and the lower panel displays the average number of planets owned by each host star, the number shown is the number of samples. The red square is type A, the blue triangle is type F, the purple star is type G, the green hexagon is type K, and the black quadrangle is type M. The gray area on the left panel is M-type stars with planets that have masses larger than 100 $M_\oplus$, and their average mass and metallicity are displayed on the black dot on the right panel (the red error bar is the standard deviation). The narrow orange area in the left panel corresponds to the right panel, which shows the cutoff mass of the K-type host star when its metal abundance jumps, which is about 0.8 $M_\odot$.

According to previous studies [23], ice giants with a mass close to Neptune are relatively common around small stars, while gas giants with a mass close to Jupiter are relatively rare. As can be seen from the left panel in Figure 8, stars with smaller masses tend to have planets with smaller masses, and F-type stars tend to have planets with larger masses than GK-type stars. However, on average, each F-type star has fewer planets than each GK-type star (bottom-right panel). The rapid rotation of F stars and a small number of narrow lines in their spectra make it difficult to detect small-mass planets in F stars, which leads to the result of a selection effect, but it cannot be ruled out that there are still many small-mass stars in F stars that have not been detected. For the transit method, people tend to look for planets in star systems with high metallicity and nearby bright sun-like star systems, so there are more investigations into G-type stars, which may also reduce the probability of finding planets in F-type stars. Additionally, low-mass K-type stars tend to have smaller metal abundances, and the metal abundance is truncated at about 0.8 $M_\odot$, but there is no similar phenomenon in the distribution of the relationship between the host stars and the planets in mass. This may means that, compared to FG-type stars, K-type stars need a higher metal abundance to form planets with a larger mass [45]. Interestingly, the behavior of M-type stars is different from that of GK-type stars. Among the M-type host stars with planets with a mass greater than 100 Mo, the average host star mass is 0.519 $M_\odot$, but the average metallicity reaches 0.158 *dex*, and a mall-mass and metal-rich M-type stars have high planetary ownership rates and can also have massive planets, which suggests that the formation mechanism of planets around M-type stars is different from that of

those around GK-type stars. This observation further supports the notion that higher metal abundance is a condition or even a prerequisite for the formation of massive planets [45].

### 3. Feasibility Analysis of FAST Radio Observation of Extrasolar Planets and Their Host Stars

FAST, with its 500 m diameter, is the largest single-antenna radio telescope in the world, renowned for its precision and sensitivity [46]. One of its crucial scientific objectives is the SETI project, which aims to detect extraterrestrial civilizations through radio observations [47]. Unlike pulsars, which have strong magnetic fields, stars and planets possess weaker magnetic fields and lower radio emission flux, making them ideal candidates for observation using high-resolution radio telescopes like FAST [48]. To this end, we rely on the sensitivity limit of FAST in different frequencies obtained through calculations by [49], with relevant parameters detailed in Table 3.

**Table 3.** FAST characteristics at frequencies below L-band.

| Frequency (MHz) | $T_{sky}$ (K) | $T_{sys}$ (K) | Sensitivity (mJy) | Angular Resolution (arcmin) |
|---|---|---|---|---|
| 70 | 2500 | 100 | 0.5 | 50 |
| 140 | 420 | 80 | 0.1 | 25 |
| 280 | 72 | 40 | 0.02 | 13 |
| 560 | 13 | 10 | 0.004 | 7 |
| 1150 | 5 | 10 | 0.003 | 3 |

Note. $T_{sky}$ is the antenna temperature, and $T_{sys}$ is the system temperature.

*Radio Flux Estimation of Planets and Their Host Stars*

Both thermal and nonthermal radiation are the main sources of radio emission from planets and their host stars. These emissions can be both coherent and incoherent. Incoherent radiation mechanisms include bremsstrahlung radiation and cyclotron synchrotron radiation, whereas coherent radiation mainly includes cyclotron radiation and plasma radiation [50]. However, it is the ionized wind produced by massive stars due to mass loss that mainly leads to continuous radio emission [51,52]. When the stellar wind emitted by a star interacts with the magnetic field of a planet, it forms a bow shock wave due to the supersonic motion of the wind. This shock wave accelerates electrons, resulting in the generation of non-thermal radiation across a wide wavelength range [53]. To measure these emissions, we utilized the simple and practical Bode's law of radiation measurement, as suggested by [26].

[26] developed a simplified model for the characteristic emission frequency and radio flux density of exoplanets by combining previous models [54–57]. According to this model, the characteristic emission frequency of planets can be expressed as:

$$V_p \sim 23.5 MHz \left(\frac{\omega}{\omega_J}\right)\left(\frac{M_p}{M_J}\right)^{\frac{5}{3}}\left(\frac{R_p}{R_J}\right)^{-3} \qquad (8)$$

Similarly, the radio flux density of exoplanets is given by:

$$S_v \sim 4.6 mJy \left(\frac{\omega}{\omega_J}\right)^{-0.2}\left(\frac{M_p}{M_J}\right)^{-0.33}\left(\frac{R_p}{R_J}\right)^{-3}\left(\frac{\Omega}{1.6sr}\right)^{-1}\left(\frac{d}{10pc}\right)^{-2}$$
$$\left(\frac{a}{1au}\right)^{-1.6}\left(\frac{\dot{M}_{ion}}{10^{-11}M_\odot yr^{-1}}\right)^{0.8}\left(\frac{v_\infty}{100kms^{-1}}\right)^2 \qquad (9)$$

In Equations (8) and (9), $\omega$ represents the rotation speed of the planet, $\omega_J$ is the rotation speed of Jupiter, $M_p$ is the planet mass, $M_J$ is the Jupiter mass, $R_p$ is the planet radius, $R_J$ is the Jupiter radius, $\Omega$ is the radio emission angle, $d$ is the distance from the Earth, $a$ is the orbital semi-major axis of the planet, $\dot{M}_{ion}$ is the mass loss rate of the star, and $v_\infty$ is the

final speed of the stellar wind.

In most observed data, the semi-major axes of exoplanets are less than 0.1 AU, and we assume that these planets are tidally locked, meaning that their rotation periods are the same as those for their orbits. The radio emission angle of a planet is compared to that of Jupiter, which is determined by the ratio of their volumes: $\Omega/\Omega_{Jupiter} = V/V_{Jupiter}$. The stellar wind speed and mass loss rate of the host star can be estimated using formulas proposed by [57] and observations from [58]. We can obtain the time dependence of the stellar wind speed using the following equations [26]:

$$v_\infty = 0.75 v_{esc} = 0.75 \times 617.5 \sqrt{\frac{R_\odot M_\star}{R_\star M_\odot}} \tag{10}$$

The particle density can be determined to be [57]:

$$n(1AU, t) = n_0 (1 + \frac{t}{\tau})^{-1.86 \pm 0.6} \tag{11}$$

$$n(d) \propto d^{-2} \tag{12}$$

For the nominal case, $v_{esc}$ is the photospheric escape velocity and $n_0 = 1.04 \times 10^{11}$ m$^{-3}$. The time constant is $\tau = 2.56 \times 10^7$ yr, calculated from [58], and $t$ is the age of the star. From these quantities, the stellar mass loss rate $\dot{M}_{ion}$ can be calculated using the following formula:

$$\dot{M}_{ion} = 4\pi d^2 n(d) v_\infty m (\frac{R_\star}{R_\odot})^2 \tag{13}$$

Here, $d$ is the distance to the host star, $m$ is the mass of hydrogen atoms in the stellar wind, $R_\star$ is the host star radius, and $R_\odot$ is the solar radius. With these parameters, we can predict the radio emission of a planet with satisfactory precision.

We selected the most recent exoplanet samples based on their semi-major axis, orbital period, radius, and mass, as well as the host star's mass, distance, spectral type, position measurements, and age. We restricted the relative error of the planet's true mass to 20%, and other parameters to 10%. We also excluded host stars with ages less than 0.7 Gyr and of M type, resulting in 212 available samples. According to [59], all objects with a mass greater than 0.58 $M_\odot$ or brightness greater than 103.6 $L_\odot$ show direct characteristics of wind in their spectra. Taking into account the observational limitations of FAST's sky area ($-14.4$–$65.6$ DEC), we estimated the radio emission frequency and flux density, which are shown in Figure 9.

In Figure 9, the x axis represents the predicted planetary radio transmission frequency and the y axis represents the predicted planetary radio flux density. The blue line represents the approximate sensitivity of the FAST (for an integration time of 1 h and a bandwidth of 4 MHz), based on the data from Table 3 of [49]. The gray-shaded area indicates the ionospheric cutoff area, where frequencies below 10 MHz cannot be observed from the ground. The yellow band represents the observation band (30 MHz–3 GHz) that FAST is capable of observing in the future. The red triangles represent planets within the observable sky range of FAST, while the blue triangles represent planets outside of that range.

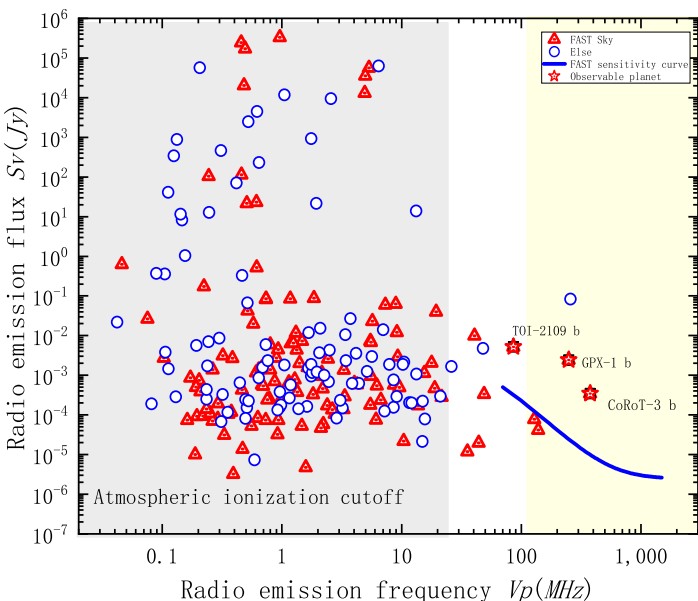

**Figure 9.** Estimated radio flux density vs. frequency of exoplanets. The red triangles represent the planets located within the FAST sky, while the blue circles represent planets outside the FAST sky. The blue cutoff line represents the minimum flux cutoff line calculated by [49]. The gray area shows the observation cutoff curve caused by atmospheric ionization, and the yellow area represents the frequency band observed by FAST.

Our results show that there are three planets in the FAST sky that may be detected by FAST, namely CoRoT-3 b, GPX-1 b and TOI-2109 b. We show the physical parameters of them and their host stars in Table 4. And, our results also indicate that most predicted planetary radio emissions fall in the low-frequency region and that the cutoff frequency of atmospheric ionization is 10 MHz, making it difficult to directly observe exoplanets with radio telescopes on the ground. The radio emission flux density of exoplanets decreases as the emission frequency increases, and data points with low emission frequency and low radiation flux density are mainly associated with small-mass planets that cannot be detected by current radio detection methods. This is consistent with the trend predicted by [49,60]. Compared to the model proposed by [49], our prediction is characterized by a wider range of the predicted flux distribution at lower frequencies. One possible explanation for this discrepancy is that the model of [49] assumes a fixed radio emission angle for exoplanets, equivalent to that of Jupiter, while our model uses the volume ratio of exoplanets to Jupiter to predict their radiation angle, leading to larger predicted radiation for larger exoplanets and smaller radiation for smaller exoplanets. Additionally, our model only accounts for the radio emission driven by the stellar wind load of the exoplanet magnetosphere, while ignoring a weak emission originating from the exoplanet itself. On the other hand, our model assumes that the exoplanet is tidally locked, resulting in an underestimation of its rotation and thus smaller predicted radiation. Our prediction can provide a lower limit of the expected exoplanet radiation. It is worth noting that the recent discovery of high circular polarization radio emission by the LOFAR telescope at low frequencies (30MHz) reported by [61] suggests that, with ongoing advancements in technology and equipment, scientists may be able to directly observe an increasing number of radio signals emitted by exoplanets.

**Table 4.** Physical parameters of CoRoT-3 b, GPX-1 b, and TOI-2109 b, and their host stars.

| Pl_Name | Pl_Orbper | Pl_Orbsmax | Pl_Radj | Pl_Massj | $V_p$ (MHz) | $S_v$ (Jy) |
|---|---|---|---|---|---|---|
| CoRoT-3 b | 4.2567994 | 0.05694 | 0.993 | 21.23 | 376.2700347 | 0.000346904 |
| GPX-1 b | 1.744579 | 0.0338 | 1.47 | 19.7 | 249.8321386 | 0.002437414 |
| TOI-2109 b | 0.67247414 | 0.01791 | 1.347 | 5.02 | 86.27965369 | 0.00521114 |
| hostname | st_teff | st_mass | sy_dist | st_spectype | st_rad | st_age (Gyr) |
| CoRoT-3 | | 1.36 | 768.771 | F3 V | 1.54 | 2.2 |
| GPX-1 | 7000 | 1.68 | 655.013 | F2 | 1.56 | 0.27 |
| TOI-2109 | 6540 | 1.45 | 262.041 | F | 1.7 | 1.77 |

**Note**. Where pl_orbper is the orbital period (days), pl_orbsmax is the semi-major axis (au), pl_radj is the planet radius (Jupiter radius), pl_massj is the planet mass (Jupiter mass), $V_p$ is the predicted emission frequency (mHz), $S_v$ is the predicted emission flux (Jy), hostname is the host star st_teff is surface temperature of the host star, st_mass is the host star mass ($M_\odot$), sy_dist is the distance (pc), st_spectype is the host star spectral type, st_rad is the host star radius ($R_\odot$), and st_age is the host star age (Gyr).

## 4. Discussion and Summary

In this paper, we analyzed data from more than 4900 confirmed exoplanet systems as of 24 February 2022. A total of 32,112 samples containing duplicates were used to investigate statistical distribution analysis of exoplanet and host star properties, as well as their relationships.

While the true mass and radius distribution of planets remains relatively unchanged, the current exploration strategy limits the discovery of small and medium-mass planets located far away from their host star. However, by breaking and optimizing the exploration strategy, we can expect to discover more small and medium-mass planets in the future. With the increase in sample sizes, we may also observe a corresponding quality valley in the radius distribution of Neptune.

Among the various planetary parameters, we focused on the influence of planet size disturbance on eccentricity in planetary systems. Our sample suggests that planetary systems with larger radius fluctuations tend to have higher eccentric fluctuations, while systems with fewer planets usually have smaller eccentric fluctuations. This indicates that the planet size inhomogeneity within a system is likely to disturb the eccentricities of its planets. However, due to the limited number of samples, this conclusion is subject to uncertainty, and finding a sample with all planets in a known system having true mass, radius, and eccentricity measurements with small errors remains challenging.

For planets with relatively small errors in the measurements of radius, true mass, and semi-major axis of orbit, the true mass–radius relationship appears to follow a piecewise power-law distribution, with the power-law index showing a downward trend in most in situ formed races from super-Earth to cold Jupiter. We speculate that this power–law index may reflect the radius dependence of the mass or the mass dependence of the radius, which could be closely related to the atmospheric proportion of planets.

We also revised the radius valley. With the increase samples of the planet, the location of the valley may be more accurate, which is of great significance to the formation and migration history of close-in planets and the inference of their atmosphere and core components.

Regarding mass parameters between host stars and planets, we notice that there may be a positive correlation between the masses of GKM-type host stars and planets. Small-mass and metal-rich M-type stars have high planetary ownership rates and can also have massive planets, which suggests that the formation mechanism of planets around M-type stars is different from that around GK-type stars. K-type stars need a higher metal abundance to form planets with a larger mass [45].

Finally, we assessed the possibility of detecting exoplanets using the FAST. According to our prediction model, there are three planets in FAST sky that may be detected by FAST,

namely CoRoT-3 b, GPX-1 b, and TOI-2109 b. Estimating the interaction between the planet and its host star remains a source of uncertainty when predicting the radio emission of exoplanets. However, the radio observations of exoplanets not only provide a new method for the direct detection but also enable the measurement of the magnetic field strength and rotation period [26], expanding the parameter space and providing a rare opportunity to directly understand the internal structure of planets. With the continued development of radio observation technology, radio detection may become a promising new method for detecting exoplanets.

**Author Contributions:** Conceptualization, L.Z.; methodology, L.Z. and B.L.; investigation, B.L.; writing—review and editing, L.Z., B.L., X.L.H. and P.M.; data curation, T.S. and L.L. All authors have read and agreed to the published version of the manuscript.

**Funding:** Our research is supported by the NSFC Grant Nos. 11963002 and 12373032. We also thank Cultivation Project for FAST Scientific Payoff and Research Achievement of CAMS-CAS.

**Data Availability Statement:** The data underlying this article are available in ChinaVO PaperData Repository, at https://doi.org/10.12149/101319.

**Conflicts of Interest:** The authors declare no conflict of interest.

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
