# Peer review of "Statistical and Radio Analysis of Exoplanets and Their Host Stars"

_universe, doi:10.3390/universe9110475_

Round 1
Reviewer 1 Report
Comments and Suggestions for Authors
The manuscript is devoted to the analysis of some observed statistical regularities and properties of exoplanet systems.
In study: (i) Distribution of exoplanets by radii (Section 2.3.1); (ii) Dependence of eccentricities of exoplanets' orbits on their radii and the number of planets in the system (i.e., orbiting a host star) (Sec. 2.3.2) ; (iii) Approximation of the mass-radius dependence for exoplanets spread to five different types (super-Earths, sub-Neptunes, sub-Saturns, cold Jupiters, hot Jupiters) (Sec. 2.3.3); (iv) Position of the " radius valley," the position of which is refined to 1.7-1.9 Earth radii (Sec. 2.3.4); (v) Dependence of the planets' masses on the mass of the parent star (Sec. 2.4); (vi) Calculations of power and frequency of radio emission of known exoplanets, caused by interaction of planets' magnetospheres with stellar winds. The study of the possibility of registration of this radio emission by the FAST radio telescope is carried out. (Sec. 3).
The scope of this study is relevant and raises interesting and important issues, but the presentation contains a number of severe drawbacks. The authors study the observed distributions, neglecting the effects of observational selection and not taking them into account in any way. Thus, the authors consider only planets with well-defined masses and radii (less than 10% error). Sometimes they relax the accuracy requirements to 50%. In other words, the authors consider only transiting exoplanets with measured masses. This raw estimation contradicts in accuracy with concluded items.
(Major)
1. Missing to take into account the observational selection leads to huge systematic errors. For example, Figure 3 shows the distribution of the planets by radii. In the region of 0.6-0.9 Jupiter radii there is a deep minimum, which the authors manifest as the desert of sub-Saturns. However, the presented distribution is the sum of the distributions of the planets detected by the space missions (Kepler, TESS) and the planets detected by the ground-based transit surveys. Ground-based surveys are not sensitive to planets with radii less than 0.7-0.8 Jupiter radii due to the influence of the Earth's atmosphere. On the distribution of planets detected solely by Kepler, the desert of sub-Saturns is almost invisible:

Distribution by radius of planets discovered by Kepler (red line) and planets discovered by ground surveys (blue line).
2. Similarly, Figure 2 shows the observed distribution of exoplanets by mass, not the “true” unbiased distribution. According to the shown graph, most the exoplanets have ~2 Jupiter masses, which is obviously not the true case. On the contrary, according to Kepler data, the number of Neptunes is an order of magnitude greater than the number of gas giants.
3. In Sec. 2.3.2, the authors study the dependence of variations of eccentricities of orbits in multiplanet systems on variations in the sizes of the planets. However, the choice of axes remains unclear. It is not clear from the text what Rp_Average and Ep_Average are. Is the averaging performed on the planets of one system or on all planets?
4. By limiting themselves to planets with known masses, radii and eccentricities, the authors consider only 81 planets in 28 systems out of over 4900 planets. A small sample can lead to both systematic and statistical errors.
5. In Sec. 2.3.3 it is not clear how the authors divided the planets into types. In Figure 6, the distribution of planets looks continuous. How do sub-Neptunes differ from sub-Saturns, and those from hot Jupiters? The boundaries of the ranges and the criteria for assigning the planets to one or another group are not given.
6. In Sec. 2.3.4, the authors explore the " radius valley ". They introduce calibrated radii (formulas 6, 7), but do not indicate how the values of coefficients g, h were obtained. Why g = -0.09, h = 0.26?
7. In Figure 8, the depth of the radius valley appears to be approximately the same for the distribution over the normal radii (g = h = 0, blue line) and over the calibrated radii (g = -0.09, h = 0.26, green line). So the necessity of introducing calibrated radii is not clear. The number of histogram bins (N = 60) is too large, it leads to increasing statistical error, the histograms do not look smooth. According to Sturges' rule (Sturges H. (1926). The choice of a class-interval. J. Amer. Statist. Assoc., 21, 65-66.) for 1213 planets the optimal number of bins N = 11. As a result, the authors' conclusion about the refinement of the position of the valley of radii does not seem somehow reasonable.
8. In Section 2.4, the authors show that as the mass of the parent star increases, the mass of the planets increases too. Again, however, authors do not take into account observational selection. The rapid rotation of F stars and the small number of narrow lines in their spectra make it just impossible to detect planets of small masses in F stars. Therefore, the discovered planets are indeed massive, but this says nothing about the possible presence of light planets in F stars.
9. Sec. 3 looks interesting and almost relevant, although it leads to disappointing conclusions about the impossibility of recording radio emissions from most of the known exoplanets with FAST.
(Minor)
1. Lines 26-27. Strictly speaking, the first exoplanets were discovered in 1992 near the pulsar PSR B1257+12 (Wolszczan, 1994). Mayor and Queloz discovered the first planet in a normal (non-neutron) star.
2. Lines 43-44. The dependence of mass on the radius of exoplanets cannot be described by a single power law due to differences in the chemical composition of exoplanets. At best, it can be described by a piecewise power law, see for example (Chen and Kipping, 2017).
3. Line 45. The radius valley also includes planets with periods of less than 3 days. Instead of "periods between 3 and 30 days," it would be better to write "periods less than 30 days."
4. Lines 68-69. The site is called the NASA Exoplanet Archive. There are other directories, such as The Extrasolar Planets Encyclopaedia.
5. Lines 147-149. The sub-Saturn desert is a seeming feature resulting from not accounting for observational selection, see major comments.
6. Line 218. A typo, correctly 0.056 AU.
7. Lines 230-233. "main star" is usually spelled host star.
8. Line 251. A typo, M < 0.45 solar masses.

Author Response
The manuscript is devoted to the analysis of some observed statistical regularities and properties
of exoplanet systems. In study: (i) Distribution of exoplanets by radii (Section 2.3.1); (ii) Dependence of eccentricities of exoplanets' orbits on their radii and the number of planets in the system (i.e., orbiting a host star) (Sec. 2.3.2) ; (iii) Approximation of the mass-radius dependence for exoplanets spread to five different types (super-Earths, sub-Neptunes, sub-Saturns, cold Jupiters, hot Jupiters) (Sec. 2.3.3); (iv) Position of the " radius valley," the position of which is refined to 1.7-1.9 Earth radii (Sec. 2.3.4); (v) Dependence of the planets' masses on the mass of the parent star (Sec. 2.4); (vi) Calculations of power and frequency of radio emission of known exoplanets, caused by interaction of planets' magnetospheres with stellar winds. The study of the possibility of registration of this radio emission by the FAST radio telescope is carried out. (Sec. 3).
The scope of this study is relevant and raises interesting and important issues, but the
presentation contains a number of severe drawbacks. The authors study the observed distributions, neglecting the effects of observational selection and not taking them into account in any way. Thus, the authors consider only planets with well-defined masses and radii (less than 10% error). Sometimes they relax the accuracy requirements to 50%. In other words, the authors
consider only transiting exoplanets with measured masses. This raw estimation contradicts in accuracy with concluded items.
Dear Editor and referees
Thank you for your perfect comments and suggestions on our manuscript entitled “Statistical and radio analysis of exoplanets and their host stars”. We agreed with all your comments. We tried our best to revise all of them. Here we send you the revised form of our manuscript. Please reconsider our manuscript again. We have modified the manuscript accordingly, and detailed corrections are listed below point by point.
(Major)
- Missing to take into account the observational selection leads to huge systematic errors. For
example, Figure 3 shows the distribution of the planets by radii. In the region of 0.6-0.9 Jupiter radii there is a deep minimum, which the authors manifest as the desert of sub-Saturns. However, the presented distribution is the sum of the distributions of the planets detected by the space missions (Kepler, TESS) and the planets detected by the ground-based transit surveys. Groundbased surveys are not sensitive to planets with radii less than 0.7-0.8 Jupiter radii due to the influence of the Earth's atmosphere. On the distribution of planets detected solely by Kepler, the desert of sub-Saturns is almost invisible:Distribution by radius of planets discovered by Kepler (red line) and planets discovered by ground surveys (blue line).
Thank you very much for your opinion. We have made relevant statistics on the radius distribution of planets again, and distinguished the planet samples of space and ground facilities. As you agree, the sub-Saturn desert is invisible in the ground search, because the ground facilities are affected by the atmosphere, and the sensitivity to planets with a radius less than 8 M⊕ is reduced. The sub-Saturn desert can be clearly seen in space surveys. Space facilities are more sensitive to the search with small mass and small radius, so the super earth and sub-Neptune with typical radius were found in a large number of space surveys, and a small valley between the super earth and sub-Neptune separated the two populations. However, due to the limitation of aperture size, the sensitivity of space observation facilities to planets with large radius and mass has decreased, but small valleys with a radius of about 8 M⊕ can still be noticed. At your suggestion, we re-described the distribution of radius.
Among them, we excluded some samples of search methods that found fewer planets, which accounted for less than 1% of the total number of samples. In the figure, the abscissa is the logarithm of radius, and the ordinate is the number of planets in each box with logarithm of radius. Among them, different color bars are used to distinguish the samples of space, ground and different discovery methods, and at the same time, we mark the radius classification of each planet population. We re-discussed the result:
Lines 123-154 :
“At the same time, because the sensitivity of ground detection and space detection to different planets is different, we distinguish space samples and ground samples, and analyze the true mass (Figure 1) and radius distribution (Figure 2) of planets according to different planetary search methods, and draw the logarithmic scale. Due to the influence of the earth's atmospheric disturbance, the sensitivity of ground facilities to the population of small-mass and small-radius planets is seriously reduced, so in the distribution of ground samples, small-mass and small-radius planets are rarely seen.Ground facilities are more sensitive to planets with a mass of about 300 M⊕ and a radius of about 15 R⊕ (about Jupiter). On the contrary, due to the vacuum in space, space facilities have a better line of sight, so they are more sensitive to small-mass and small-radius planets (the peak of mass distribution is about 9 M⊕, but due to the limited aperture of space facilities, the sensitivity to large-mass planets has declined. The mass samples in space indicate the shortage of planets near log10(M⊕) ranging from 1.1 to 2.1, with a medium mass range of 10-100 M⊕ and a radius of 6 R⊕) (the logarithmic value is 0.8), which is the statistical characteristic of the sub-Saturn desert. Due to the limited sensitivity of ground facilities, we can't find it in ground samples. In the radius distribution of space samples, planets are concentrated around the radius of 1.5 R⊕) (logarithmic value of 0.2) and 3 R⊕) (logarithmic value of 0.5), which is the typical radius of super earth and sub-Neptune. A small valley between super-Earths and sub-Neptunes separates the two populations and will become more evident with the increase of samples. The valley may indicate different formation and evolution mechanisms for sub-Neptunes and super-Earths. Small planets are commonly formed in protoplanetary disks, but only a small fraction of them grow to a critical size over time and become gas giants. Moreover, low-mass planets at close distances may experience significant light evaporation and lose a considerable portion of their original H/He inventory, resulting in a smaller radius (Lopez et al. 2012; Lopez and Fortney 2013; Owen and Wu 2013). This causes planets to accumulate at small and medium radii. i. Howard et al. (2010) predicted that 23% of stars have a planet with a mass similar to Earth's (0.5 to 2.0 M⊕). Among different search methods, transit method is undoubtedly the most successful, because it is not limited by facilities and finds most of the planets in the total sample, especially in space survey, while other search methods are mainly in ground survey, such as RV method. Imaging method is very sensitive to massive planets in wide orbits; Therefore, most of the planets detected by this technology have large radius and high mass.”
- Similarly, Figure 2 shows the observed distribution of exoplanets by mass, not the “true” unbiased distribution. According to the shown graph, most the exoplanets have ~2 Jupiter masses, which is obviously not the true case. On the contrary, according to Kepler data, the number of Neptunes is an order of magnitude greater than the number of gas giants.
Thank you very much for your suggestions. Similarly, at your suggestion, we have re-made statistics on the mass distribution of planets, distinguished the planet samples of space facilities and ground facilities, and re-discussed the relevant results. Similarly, in the ground transit sample, the sub-Saturn desert is invisible, but it is visible in the space transit sample, which is also caused by the different sensitivity of different facilities to planets of different “sizes”.
Likewise, we excluded some samples of search methods that found fewer planets, which accounted for less than 1% of the total number of samples. In the figure, the abscissa is the logarithm of mass of planets, and the ordinate is the number of planets in each box with logarithm of mass. Among them, different color bars are used to distinguish the samples of space, ground and different discovery methods. We re-discussed the results of mass distribution and radius distribution:
Lines 123-154 :
“At the same time, because the sensitivity of ground detection and space detection to different planets is different, we distinguish space samples and ground samples, and analyze the true mass (Figure 1) and radius distribution (Figure 2) of planets according to different planetary search methods, and draw the logarithmic scale. Due to the influence of the earth's atmospheric disturbance, the sensitivity of ground facilities to the population of small-mass and small-radius planets is seriously reduced, so in the distribution of ground samples, small-mass and small-radius planets are rarely seen.Ground facilities are more sensitive to planets with a mass of about 300 M⊕ and a radius of about 15 R⊕ (about Jupiter). On the contrary, due to the vacuum in space, space facilities have a better line of sight, so they are more sensitive to small-mass and small-radius planets (the peak of mass distribution is about 9 M⊕, but due to the limited aperture of space facilities, the sensitivity to large-mass planets has declined. The mass samples in space indicate the shortage of planets near log10(M⊕) ranging from 1.1 to 2.1, with a medium mass range of 10-100 M⊕ and a radius of 6 R⊕) (the logarithmic value is 0.8), which is the statistical characteristic of the sub-Saturn desert. Due to the limited sensitivity of ground facilities, we can't find it in ground samples. In the radius distribution of space samples, planets are concentrated around the radius of 1.5 R⊕) (logarithmic value of 0.2) and 3 R⊕) (logarithmic value of 0.5), which is the typical radius of super earth and sub-Neptune. A small valley between super-Earths and sub-Neptunes separates the two populations and will become more evident with the increase of samples. The valley may indicate different formation and evolution mechanisms for sub-Neptunes and super-Earths. Small planets are commonly formed in protoplanetary disks, but only a small fraction of them grow to a critical size over time and become gas giants. Moreover, low-mass planets at close distances may experience significant light evaporation and lose a considerable portion of their original H/He inventory, resulting in a smaller radius (Lopez et al. 2012; Lopez and Fortney 2013; Owen and Wu 2013). This causes planets to accumulate at small and medium radii. i. Howard et al. (2010) predicted that 23% of stars have a planet with a mass similar to Earth's (0.5 to 2.0 M⊕). Among different search methods, transit method is undoubtedly the most successful, because it is not limited by facilities and finds most of the planets in the total sample, especially in space survey, while other search methods are mainly in ground survey, such as RV method. Imaging method is very sensitive to massive planets in wide orbits; Therefore, most of the planets detected by this technology have large radius and high mass.”
- In Sec. 2.3.2, the authors study the dependence of variations of eccentricities of orbits in multiplanet systems on variations in the sizes of the planets. However, the choice of axes remains unclear. It is not clear from the text what Rp_Average and Ep_Average are. Is the averaging performed on the planets of one system or on all planets?
Thank you for your suggestion. We have added the explanations of Rp_Average and Ep_Average. Rp_Average is the average of all planetary radii in the same system, while Ep_Average is the average of all planetary eccentricities in the same system.
- By limiting themselves to planets with known masses, radii and eccentricities, the authors consider only 81 planets in 28 systems out of over 4900 planets. A small sample can lead to both systematic and statistical errors.
Thank you for your suggestions. The selected samples are few for the whole sample, but we suppose that this may be a direction that we can pay attention to, that is, whether the eccentricity of the planets in the multi-planetary system may be disturbed by the larger-scale planets, which requires more perfect data samples to explore this point. We relaxed the relative error limit and obtained more complete parameter data by crossing all the recorded parameters. We expanded our sample (90% of these samples are space transit samples): 42 systems with two planets, 23 with three planets, 9 with four planets, 3 with five planets, and 3 systems (Kepler-11, K2-138, and TRAPPIST-1) with six and seven planets}, respectively (without any restrictions on the host star).
We re-discussed the results accordingly:
Lines 173-186 :
“Larger planets need more gravity to restrain them. If there is a planet with a larger scale in the system, will it have a greater impact on the orbit of the whole system? To quantify the radius and eccentricity fluctuations in each system, we used the logarithm (Rp_Max) - logarithm (Rp_Average) and logarithm (Ep_{Max) - logarithm (Ep_Average) values, respectively. Rp_Average is the average radius of all planets in the same system, while Ep_{Average} is the average eccentricity of all planets in the same system, and Rp_{Max} and Ep_{Max} respectively represent the maximum radius and eccentricity of planets in the system. As shown in Figure 3, the larger fluctuation of planet ``size'' is often related to the higher eccentric fluctuation in the system, and the system with fewer planets tends to show smaller eccentric fluctuation on average. The uneven distribution of planet ``size'' may impact the stability of planetary orbits. Although the selected samples are few for the whole sample, which may cause large errors, we consider that this may be a direction that can be of concern, that is, whether the eccentricity of the planets in the multi-planetary system may be disturbed by the larger-scale planets, which requires more complete data samples to discuss this.”
- In Sec. 2.3.3 it is not clear how the authors divided the planets into types. In Figure 6, the distribution of planets looks continuous. How do sub-Neptunes differ from sub-Saturns, and those from hot Jupiters? The boundaries of the ranges and the criteria for assigning the planets to one or another group are not given.
Thank you for your suggestions. We added the classification of planetary population in the table, and the classification is the same as that of the population on the radius.
- In Sec. 2.3.4, the authors explore the " radius valley ". They introduce calibrated radii (formulas 6, 7), but do not indicate how the values of coefficients g, h were obtained. Why g = -0.09, h = 0.26?
Thank you for your suggestions. We have added a discussion that quoted the calibration radius and correlation coefficients g and h. According to previous reports, the radius position of the valley decreases with the increase of orbital period (Van Eylen et al. 2018) and increases with the increase of the host star mass (Berger et al. 2020a). When the orbital period P = 10 days and the host mass M = M☉, the slope quantifying the period dependence is g = (VanEylen et al. 2018). The slope quantifying the stellar mass dependence is h = (Berger et al.2020a). By quantifying the dependence of the period and the mass of the host star, we can obtain the calibration radius that can highlight the radius valley better.
- In Figure 8, the depth of the radius valley appears to be approximately the same for the
distribution over the normal radii (g = h = 0, blue line) and over the calibrated radii (g = -0.09, h =
0.26, green line). So the necessity of introducing calibrated radii is not clear. The number of
histogram bins (N = 60) is too large, it leads to increasing statistical error, the histograms do not
look smooth. According to Sturges' rule (Sturges H. (1926). The choice of a class-interval. J. Amer.
Statist. Assoc., 21, 65-66.) for 1213 planets the optimal number of bins N = 11. As a result, the
authors' conclusion about the refinement of the position of the valley of radii does not seem somehow reasonable.
Thank you for your suggestions. We have adjusted the transparency of the statistical graph in Figure 7 to make the three different methods more distinguishable. It can be noted that the calibrated radius valley (red, green) is deeper (about 0.005 ratio) than the uncalibrated radius (blue), and the valley moves forward by about 0.1R⊕. The histogram box here represents the ratio of the number of planets in each 0.1 step radius ranging from 0 to 30 R⊕ to the total number of stars in the sample system, and we have added the explanation of the histogram box. In order to better show the radius valley, we only show the results of 0-6 R⊕.
- In Section 2.4, the authors show that as the mass of the parent star increases, the mass of the
planets increases too. Again, however, authors do not take into account observational selection.
The rapid rotation of F stars and the small number of narrow lines in their spectra make it just
impossible to detect planets of small masses in F stars. Therefore, the discovered planets are
indeed massive, but this says nothing about the possible presence of light planets in F stars.
Thank you for your suggestions. We rewritten the discussion. We added your comments, discussed the F stars separately, and excluded the F stars from the comparison.
Lines : 358-328
“The rapid rotation of F stars and a small number of narrow lines in their spectra make it difficult to detect small-mass planets in F stars, which leads to the result of a selection effect, but it cannot be ruled out that there are still many small-mass stars in F stars that have not been detected.”
- Sec. 3 looks interesting and almost relevant, although it leads to disappointing conclusions
about the impossibility of recording radio emissions from most of the known exoplanets with
FAST.
Thank you for your comments!
(Minor)
- Lines 26-27. Strictly speaking, the first exoplanets were discovered in 1992 near the pulsar PSR
B1257+12 (Wolszczan, 1994). Mayor and Queloz discovered the first planet in a normal (nonneutron) star.
Thank you for your suggestions! We added (nonneutron) as a reminder.
Line 88
- Lines 43-44. The dependence of mass on the radius of exoplanets cannot be described by a
single power law due to differences in the chemical composition of exoplanets. At best, it can be
described by a piecewise power law, see for example (Chen and Kipping, 2017).
Thank you for your suggestions! We made a modification and added "piecewise" to prevent confusion.
- Line 45. The radius valley also includes planets with periods of less than 3 days. Instead of
"periods between 3 and 30 days," it would be better to write "periods less than 30 days."
Thank you for your suggestions! We have changed "periods between 3 and 30 days," to "periods less than 30 days."
- Lines 68-69. The site is called the NASA Exoplanet Archive. There are other directories, such as
The Extrasolar Planets Encyclopaedia.
Thank you for your suggestions! We modified this point and added a note.
Lines:69
“The NASA Exoplanet Archive1 provides basic information on exoplanets and their host stars”
- Lines 147-149. The sub-Saturn desert is a seeming feature resulting from not accounting for
observational selection, see major comments.
Thank you for your suggestions! We have adjusted the sample and explained that this is a statistical feature.
Line: 132-135
“The mass samples in space indicate the shortage of planets near log10(M⊕) ranging from 1.1 to 2.1, with a medium mass range of 10-100 M⊕ and a radius of 6 R⊕) (the logarithmic value is 0.8), which is the statistical characteristic of the sub-Saturn desert. Due to the limited sensitivity of ground facilities, we can't find it in ground samples.”
- Line 218. A typo, correctly 0.056 AU.
Thank you for your suggestions! We have modified it.
Lines:227
“0.056”
- Lines 230-233. "main star" is usually spelled host star.
Thank you for your suggestions! ,We have modified the “main star” in the paper to “host star”.
- Line 251. A typo, M < 0.45 solar masses.
Thank you for your suggestions! ,We have modified the “M⊕ ” in the paper to “M☉”.
Lines :260
We revised them to “M☉”
Thank you for your hard work again. We tried our best to revise all of them. It is important for Li Baoda to use the manuscript to apply for the Master degree.
Best Wishes
Zhang liyun
Guizhou University

Reviewer 2 Report
Comments and Suggestions for Authors
With the launch of the TESS space telescope and the increasing accuracy of radial velocity detection using the ground-based high-resolution spectrograph, the number of exoplanets confirmed will become large. It is very important to analyze the distribution of exoplanets based on large sample dataset, and then to explore the theory of the formation and evolution of exoplanets. The authors selected the samples in the latest and largest dataset according to their own criterion to analyze the distribution of the parameters of the exoplanets and the host stars, and made a preliminary exploration on the detection possibility of the largest radio telescope in the world (FAST) for exoplanets. Although the sample is small, the methodology and new conclusions are interesting.
However, some of the sample selection processes in the article are too simple, and I don't understand the basis for them (for example, how to eliminate the detection method bias). In addition, some conclusions lack quantitative proof. Therefore, further revisions are required to re-evaluate whether it is worthy of publication.
Significant revisions should be done based on my following comments.
1.This is the most major comment in this review report. As the authors mentioned, different detection methods can cause the samples to have detection bias(e.g., Section 2.1 & 2.2). The authors only chose the relative error to select the samples, but did not distinguish between different detection methods. The authors only distinguish between different detection methods when plotting figures (e.g., Figure 2 & 3). However, due to the small number of discovered exoplanets, the samples of Radial Velocity and Imaging methods are not highlighted, so the figures needs to be revised. I suggest that all samples and the samples of different methods be plotted on a separate panel, or plotted on the same panel after normalization (which can be visually compared). Convince the reader clearly that the authors' analyzes are not subject to detection method bias.
2. In Figure 4, “purple stars (5+ planets)” may not satisfy the rule that “the average eccentricity tends to decrease as the number of planets in the system increases”. The authors should explain the reason.
3. In equations (6) and (7), and Figure 8, in the calculation of the “calibrated radius”, the sources and meanings of the values of “g” and “h” are not discussed in detail, and readers may be confused.
4. The description of sample selection is not clearly stated. For example, In Section 2.3.4, Line 250, “…, excluding stars 250 (M<0.45 M⊕), and allowing only one host star per system.”, the author did not discuss why this is done. The authors should discuss it. In the same case, In Section 3.1 Line 353, “also excluded main stars with age less than 0.7 Gyr and of M type,” The authors explain why this is done.
5. In Section 2.3.4, Line 267-268, “Compared to previous studies, the radius valley we reproduced is narrower and more pronounced.” Qualitative statements cannot convince readers, and quantitative calculations are required. The authors should conduct some statistical tests to check whether this conclusion is statistically reliable.
6. In Section 2.3.4, Line 275-278, it is good that the author makes this point that radius valley is important for statistical inference of the characteristics of low-mass planets. But the authors do not propose the theoretical scientific significance of the new results obtained in this work for the formation and evolution of exoplanets. Therefore, I suggest that the author give further guidance on the two formation mechanisms ("light evaporation of the atmosphere and the core-driven mass loss mechanism").
7. The figures need to be modified. In Figure 1, in addition to the number of stellar systems of each type, percentages can be added. The y-axis can be plotted on a logarithmic scale. In addition, Figure 1 is not necessary, and can be replaced by a table.
8. Many legends of the figures are very incomplete. I only use Figure 7 as an example for illustration. The caption in Figure 7 does not indicate the meanings of the regions and boxes of different shapes at all. Why the authors choose the calibrated radius from 1 to 4 R⊕ in the black box in Figure 7? In addition, it is suggested that this picture be added with a histogram on the side to make the conclusion more prominent and easier for readers to get. Other figures also have similar problems. The author should revise carefully.
9. About academic language, some academic language is not very standardized, or not unified throughout the text, which makes readers a little confused (e.g., the main stars and the host stars). The writing format in the text is also not standardized (e.g., “imf” and “rmf” in Line 203).

Author Response
Review Report
With the launch of the TESS space telescope and the increasing accuracy of radial velocity
detection using the ground-based high-resolution spectrograph, the number of exoplanets
confirmed will become large. It is very important to analyze the distribution of exoplanets
based on large sample dataset, and then to explore the theory of the formation and evolution
of exoplanets. The authors selected the samples in the latest and largest dataset according to
their own criterion to analyze the distribution of the parameters of the exoplanets and the host
stars, and made a preliminary exploration on the detection possibility of the largest radio
telescope in the world (FAST) for exoplanets. Although the sample is small, the methodology
and new conclusions are interesting. However, some of the sample selection processes in the article are too simple, and I don't understand the basis for them (for example, how to eliminate the detection method bias). In addition, some conclusions lack quantitative proof. Therefore, further revisions are required to re-evaluate whether it is worthy of publication. Significant revisions should be done based on my following comments.
Dear Editor and referees
Thank you for your perfect comments and suggestions on our manuscript entitled “Statistical and radio analysis of exoplanets and their host stars”. We agreed with all your comments. We tried our best to revise all of them. Here we send you the revised form of our manuscript. Please reconsider our manuscript again. We have modified the manuscript accordingly, and detailed corrections are listed below point by point.
1.This is the most major comment in this review report. As the authors mentioned, different detection methods can cause the samples to have detection bias(e.g., Section 2.1 & 2.2). The authors only chose the relative error to select the samples, but did not distinguish between different detection methods. The authors only distinguish between different detection methods when plotting figures (e.g., Figure 2 & 3). However, due to the small number of discovered exoplanets, the samples of Radial Velocity and Imaging methods are not highlighted, so the figures needs to be revised. I suggest that all samples and the samples of different methods be plotted on a separate panel, or plotted on the same panel after normalization (which can be visually compared). Convince the reader clearly that the authors' analyzes are not subject to detection method bias.
Thank you very much for your suggestions. Due to the influence of the earth's atmosphere, the ground survey is insensitive to planets with a radius less than 0.7-0.8 Jupiter, while the space survey is more sensitive to planets with a smaller radius, thus producing a selection effect. We have re-counted the radius and true mass distribution of the planet, and distinguished the planet samples of space and ground facilities. Among them, we excluded some samples of search methods that found fewer planets, which accounted for less than 1% of the total number of samples. In figure 1, the abscissa is the logarithm of radius, and the ordinate is the number of planets in each box with logarithm of radius. In the figure, the abscissa is the logarithm of mass of planets, and the ordinate is the number of planets in each box with logarithm of mass. Among them, different color bars are used to distinguish the samples of space, ground and different discovery methods. We re-discussed the results of mass distribution and radius distribution:
Lines 123-154 :
“At the same time, because the sensitivity of ground detection and space detection to different planets is different, we distinguish space samples and ground samples, and analyze the true mass (Figure 1) and radius distribution (Figure 2) of planets according to different planetary search methods, and draw the logarithmic scale. Due to the influence of the earth's atmospheric disturbance, the sensitivity of ground facilities to the population of small-mass and small-radius planets is seriously reduced, so in the distribution of ground samples, small-mass and small-radius planets are rarely seen.Ground facilities are more sensitive to planets with a mass of about 300 M⊕ and a radius of about 15 R⊕ (about Jupiter). On the contrary, due to the vacuum in space, space facilities have a better line of sight, so they are more sensitive to small-mass and small-radius planets (the peak of mass distribution is about 9 M⊕, but due to the limited aperture of space facilities, the sensitivity to large-mass planets has declined. The mass samples in space indicate the shortage of planets near log10(M⊕) ranging from 1.1 to 2.1, with a medium mass range of 10-100 M⊕ and a radius of 6 R⊕) (the logarithmic value is 0.8), which is the statistical characteristic of the sub-Saturn desert. Due to the limited sensitivity of ground facilities, we can't find it in ground samples. In the radius distribution of space samples, planets are concentrated around the radius of 1.5 R⊕) (logarithmic value of 0.2) and 3 R⊕) (logarithmic value of 0.5), which is the typical radius of super earth and sub-Neptune. A small valley between super-Earths and sub-Neptunes separates the two populations and will become more evident with the increase of samples. The valley may indicate different formation and evolution mechanisms for sub-Neptunes and super-Earths. Small planets are commonly formed in protoplanetary disks, but only a small fraction of them grow to a critical size over time and become gas giants. Moreover, low-mass planets at close distances may experience significant light evaporation and lose a considerable portion of their original H/He inventory, resulting in a smaller radius (Lopez et al. 2012; Lopez and Fortney 2013; Owen and Wu 2013). This causes planets to accumulate at small and medium radii. i. Howard et al. (2010) predicted that 23% of stars have a planet with a mass similar to Earth's (0.5 to 2.0 M⊕). Among different search methods, transit method is undoubtedly the most successful, because it is not limited by facilities and finds most of the planets in the total sample, especially in space survey, while other search methods are mainly in ground survey, such as RV method. Imaging method is very sensitive to massive planets in wide orbits; Therefore, most of the planets detected by this technology have large radius and high mass.”
- In Figure 4, “purple stars (5+ planets)” may not satisfy the rule that “the average eccentricity
tends to decrease as the number of planets in the system increases”. The authors should explain the reason.
Thank you very much for your suggestions. In Figure 3, "purple stars (5+planets)" represents the eccentricity fluctuation as a function of the fluctuation of the planet radius in a system with more than five planets, showing that if one planet in a planetary system is much larger than the other, the eccentricity of the planets in the system will be more disturbed. We have added the explanations of Rp_Average and Ep_Average. Rp_Average is the average radius of all planets in the same system, while Ep_{Average} is the average eccentricity of all planets in the same system, and Rp_{Max} and Ep_{Max} respectively represent the maximum radius and eccentricity of planets in the system.
Figure 4 shows the Average eccentricity in a multi-planet system, that is, the average eccentricity of samples in all planetary systems with the same number of planets, and the average eccentricity tends to decrease with the increase of the number of planets in the system. We supplemented the explanations of these two parameters and re-discussed:
Lines 173-186 :
“Larger planets need more gravity to restrain them. If there is a planet with a larger scale in the system, will it have a greater impact on the orbit of the whole system? To quantify the radius and eccentricity fluctuations in each system, we used the logarithm (Rp_Max) - logarithm (Rp_Average) and logarithm (Ep_{Max) - logarithm (Ep_Average) values, respectively. Rp_Average is the average radius of all planets in the same system, while Ep_{Average} is the average eccentricity of all planets in the same system, and Rp_{Max} and Ep_{Max} respectively represent the maximum radius and eccentricity of planets in the system. As shown in Figure 3, the larger fluctuation of planet ``size'' is often related to the higher eccentric fluctuation in the system, and the system with fewer planets tends to show smaller eccentric fluctuation on average. The uneven distribution of planet ``size'' may impact the stability of planetary orbits. Although the selected samples are few for the whole sample, which may cause large errors, we consider that this may be a direction that can be of concern, that is, whether the eccentricity of the planets in the multi-planetary system may be disturbed by the larger-scale planets, which requires more complete data samples to discuss this.”
Lines 188-190 :
We further studied the Average eccentricity of different planetary systems, as shown in Figure 5, where the average eccentricity is the average eccentricity of all samples in all planetary systems with the same number of planets.
3.In equations (6) and (7), and Figure 8, in the calculation of the “calibrated radius”, the sources and meanings of the values of “g” and “h” are not discussed in detail, and readers may be confused.
Thank you for your suggestions. We have added a discussion that quoted the calibration radius and correlation coefficients g and h.
Lines :269-275
According to previous reports, the radius position of the valley decreases with the increase of orbital period (Van Eylen et al. 2018) and increases with the increase of the host star mass (Berger et al. 2020a). When the orbital period P = 10 days and the host mass M = M☉, the slope quantifying the period dependence is g = (VanEylen et al. 2018). The slope quantifying the stellar mass dependence is h = (Berger et al.2020a). By quantifying the dependence of the period and the mass of the host star, we can obtain the calibration radius that can highlight the radius valley better.
- The description of sample selection is not clearly stated. For example, In Section 2.3.4, Line 250, “…, excluding stars 250 (M<0.45 M⊕), and allowing only one host star per system.”, the author did not discuss why this is done. The authors should discuss it. In the same case, In Section 3.1 Line 353, “also excluded main stars with age less than 0.7 Gyr and of M type,” The authors explain why this is done.
Thank you for your suggestions. We added a selection note:
Lines :262-264
Because the formation mechanism of planets around M dwarf stars may be different from that around FGK stars, we ruled it out. Meanwhile, in order to ensure that only one main star in each system can affect the planets around it, only one main star will be allowed in each system.
5.In Section 2.3.4, Line 267-268, “Compared to previous studies, the radius valley we reproduced is narrower and more pronounced.” Qualitative statements cannot convince readers, and quantitative calculations are required. The authors should conduct some statistical tests to check whether this conclusion is statistically reliable.
Thank you for your suggestions. We have adjusted the transparency of the statistical graph in Figure 7 to make the three different methods more distinguishable. It can be noted that the calibrated radius valley (red, green) is deeper (about 0.005 ratio) than the uncalibrated radius (blue), and the valley moves forward by about 0.1R⊕. The histogram box here represents the ratio of the number of planets in each 0.1 step radius ranging from 0 to 30 R⊕ to the total number of stars in the sample system, and we have added the explanation of the histogram box. In order to better show the radius valley, we only show the results of 0-6 R⊕. Compared with the previous limit of 1.9±0.2, the radius valley here is limited to 1.8±0.1, and the peak distribution on both sides of the valley is located near 1.5, that is, 2.6, which is consistent with the previous results and more accurate. We added this detail:
Lines:289-290
“Compared with the previous limit of 1.9±0.2, the radius valley here is limited to 1.8±0.1.
- In Section 2.3.4, Line 275-278, it is good that the author makes this point that radius valley is important for statistical inference of the characteristics of low-mass planets. But the authors do not propose the theoretical scientific significance of the new results obtained in this work for the formation and evolution of exoplanets. Therefore, I suggest that the author give further guidance on the two formation mechanisms ("light evaporation of the atmosphere and the core-driven mass loss mechanism").
Thank you for your suggestions. We added this part of the discussion.
Lines :295-311
Compared with the core-driven mass loss model, the valley obtained by calibration radius here is narrower, but the shapes and positions of the two peaks are close. In the explanation of the core-driven mass loss mechanism by the core, the optical envelope of the planet is optically thinner, so it cools and shrinks quickly, while the heavy envelope cools slowly, roughly maintaining its initial radius. By eroding the light atmosphere, the mass loss caused by core cooling amplifies this effect and will deepen the valley (Ginzburg et al. 2018), (Lopez2013 et al.) predicted a narrow valley in the light evaporation model, and the threshold will be around 1.8 R⊕, which is in good agreement with our results. Such a narrow occurrence valley suggests that there may not be a large number of 1-10 M⊕ rocky planets formed by huge crash after the disk has dissipated (Morbidelli et al. 2012), and it is more likely to be migrated by light evaporation or core-driven mass loss mechanism. The character (for example, location and shape) of the radius valley mainly depends on the distribution of the planet's core mass, core composition and atmospheric mass fraction (Lopez2013). If the location of the radius valley can be accurately determined, it will be helpful to determine the initial mass fraction critical value of the short-range (3-30 days) planet whose core mass is about the mass of the earth (when high-energy radiation is enough to dissolve the whole H/He atmosphere) (Owen & Wu 2017)
- The figures need to be modified. In Figure 1, in addition to the number of stellar systems of each type, percentages can be added. The y-axis can be plotted on a logarithmic scale. In addition, Figure 1 is not necessary, and can be replaced by a table.
Thank you for your suggestions. We have adjusted Figure 1 and changed it into a tabular form.
- Many legends of the figures are very incomplete. I only use Figure 7 as an example for illustration. The caption in Figure 7 does not indicate the meanings of the regions and boxes of different shapes at all. Why the authors choose the calibrated radius from 1 to 4 R⊕ in the black box in Figure 7? In addition, it is suggested that this picture be added with a histogram on the side to make the conclusion more prominent and easier for readers to get. Other figures also have similar problems. The author should revise carefully.
Thank you for your suggestions. We redraw Figure 7 and add more detailed notes, where the abscissa is the logarithm of the orbital period and the ordinate is the calibration radius. The upper diagram shows the orbital period distribution of the sample, and the right diagram shows the calibration radius distribution. The light blue bag in the middle marks the sparse area of the planet, which corresponds to the obvious valley in the calibration radius distribution on the right. The black box in the middle is a sample with a period of 3-30 days and a calibration radius of 1-4 R⊕, which will be used for statistics and drawing Figure 7.
- About academic language, some academic language is not very standardized, or not unified throughout the text, which makes readers a little confused (e.g., the main stars and the host stars). The writing format in the text is also not standardized (e.g., “imf” and “rmf” in Line203).
Thank you for your opinion. We have unified and modified it.
Thank you for your hard work again. We tried our best to revise all of them. It is important for Li Baoda to use the manuscript to apply for the Master degree.
Best Wishes
Zhang liyun
Guizhou University

Round 2
Reviewer 2 Report
Comments and Suggestions for Authors
Review Report
The article has been revised according to the previous comments, but there are still some issues that have not been explained clearly. I have three main comments for modification.
1. Comments on content:
Attach all the samples and parameters of this work in the form of a table (electronic attachments can be found on the website, and readers can download and read them).
2. Comments on the calculation process and the source of values:
(1) In Section “3.1 Radio flux estimation of planets and their host stars”, the authors do not specify the detailed calculation method. For example, regarding the calculation of radio flux density of exoplanets, Function (9) has the parameter of υ∞. The authors point out that the υ∞ is “the final speed of the stellar wind” (Line 368). As the authors mentioned in Line 369, “In most observed data, the semi-major axes of exoplanets are less than 0.1 AU ”. Function (10) only provides the calculation method for υ∞ at 1 AU (requiring stellar age). Here is my question: How do the authors calculate the υ∞ at the position of the planet? Please provide a detailed calculation process.
(2) Function (12) provides the calculation method for the mass loss rate of the star, , but the value and source of stellar wind proton are not specified in the text.
3. Comments on the samples used for analysis:
(1) In “Section 2.3.4 Planetary radius and period (radius valley)”, in Figure 7, the text in the figure shows only “Space-Transit” samples, while the text in the article mentions sample selecting (Line 258-264). but it does not indicate that only “Space-Transit” samples are left. Please explain the reason.
(2) In “Section 2.3.1. Distribution of true mass and radius of planes”, the authors analyzed the distribution of different subset samples. Why not analyze the distribution of the total sample? For example, in Figures 1 and 2, add a histogram of the total sample, and check for any changes in the conclusion about the distribution.
(3) In “Section 2.3.4 Planetary radius and period (radius valley)”, Line 260-261, the authors write "excluding stars (M<0.45 M⊕)," but in “Section 2.4 The mass of the host star and its planet”, why do the authors not exclude the low-mass M-type stars (see Figure 8)? And Figure 8 shows that there are some data points with significant errors. Please explain them.
I have some other small suggestions.
1. Figure modification:
(1) In Figure 4, the horizontal axis can be added with the error bar represented by standard deviation, to give readers credibility of the relationship trend.
(2) In Figure 8, the upper right panel and right panel are plotted separately, not within the main panel.
2. The meaning of the parameters in Table 3, such as Tsky, and Tsys.
3. Some confusing sentences:
(1)In Line 113, what does “cycle” mean?
(2)In Line 132, “… but due to the limited aperture of space facilities, the sensitivity to large-mass planets has declined.” Can the authors provide a detailed explanation of the reason?
(3)In Line 327-330, the authors only explained the bias of RV detection and it is need to add the bias of transition detection.
(4)In Figure 8, the authors mentioned “the cutoff mass of the K-type host star when its metal foundation jumps,” Please describe the causes of this phenomenon.
(5)In Line 333-334, the authors mentioned “Small-mass and metal-rich M-type stars with high planetary ownership rates and can also have mass planes,” but I cannot obtain the information on “metal-rich” from the figures or text. Please provide a detailed explanation.
(6)What is the meaning of “rows of data or exoplanets”? For example, in Line 72 and Line 426.
4. About academic language, check the written form of “lg” in Function (4) and (5); Check Line 261 “(M<0.45 M ⊕)”.

Author Response
Review Report
The article has been revised according to the previous comments, but there are still
some issues that have not been explained clearly. I have three main comments for
modification.
Dear Editor and referee1
Thank you for your perfect comments and suggestions on our manuscript entitled “Statistical and radio analysis of exoplanets and their host stars”. We agreed with all your comments. We revised all of them. Here we send you the revised form of our manuscript. Please reconsider our manuscript again. We have modified the manuscript accordingly, and detailed corrections are listed below point by point.
- Comments on content:
Attach all the samples and parameters of this work in the form of a table (electronic attachments can be found on the website, and readers can download and read them).
Thank you for your suggestions! We added a table of the samples we used in the attachment. Each table corresponds to the samples used in each section of analysis.
- Comments on the calculation process and the source of values:
(1) In Section “3.1 Radio flux estimation of planets and their host stars”, the authors do not specify the detailed calculation method. For example, regarding the calculation of radio flux density of exoplanets, Function (9) has the parameter of υ∞. The authors point out that the υ∞ is “the final speed of the stellar wind” (Line 368). As the authors mentioned in Line 369, “In most observed data, the semi-major axes of exoplanets are less than 0.1 AU ”. Function (10) only provides the calculation method for υ∞ at 1 AU (requiring stellar age). Here is my question: How do the authors calculate the υ∞ at the position of the planet? Please provide a detailed calculation process.
Thank you for your suggestions! We have noticed this problem. Except for the sun, the wind characteristics of stars are mostly unknown. Therefore, we re-use v∞ = 0.75 × vesc (where vesc is the escape velocity of the stellar photosphere) to estimate the stellar wind (E. O'Gorman),2005. The average stellar wind speed of the samples is 442.38km/s, which is similar to that of the sun. At the same time, n(d)∝d-2 and the estimated n(1Au) are used to determine the estimated stellar wind density and mass loss rate at the planetary position, so the influence of distance on the stellar wind is considered. We recalculated the radio flux that the planet might emit. Because in the previous version, we used the estimated star wind and mass loss rate normalized to 1Au, so we underestimated the emission of the planet. After correction, we found that three planets in the FAST sky area are expected to be detected by FAST, namely CoRoT-3 b, GPX-1 b and TOI-2109 b. Their physical parameters are shown in the figure below. They are similar to Jupiter, but closer to the host star.
lines:403-405
“Our results show that there are three planets in FAST sky that may be detected by FAST, namely CoRoT-3 b, GPX-1 b and TOI-2109 b. We show the physical parameters of them and their host stars in Table 4.”
(2) Function (12) provides the calculation method for the mass loss rate of the star, but the value and source of stellar wind proton are not specified in the text.
Thank you for your suggestions! The stellar wind proton is a hydrogen atom, so the value of M stellar wind proton is 1.674× 10-27 kg. We added this to the paper.
Lines:384
“m is the mass of hydrogen atoms in the stellar wind.”
- Comments on the samples used for analysis:
(1) In “Section 2.3.4 Planetary radius and period (radius valley)”, in Figure 7, the text in the figure shows only “Space-Transit” samples, while the text in the article mentions sample selecting (Line 258-264). but it does not indicate that only “Space-Transit” samples are left. Please explain the reason.
Thank you for your suggestions! We are very sorry that the label in Figure 7 was wrongly displayed as "Space-Transit". These samples are Transit samples without classification of observation facilities, not "Space-Transit" samples. Of all the 1213 samples in Figure 6, only 10 are RV samples, and the others are Transit samples, so we did not consider these 10 RV samples when calculating the planetary occurrence rate. We have corrected the “Space-Transit” in Figure 7 to “Transit”.
(2) In “Section 2.3.1. Distribution of true mass and radius of planes”, the authors analyzed the distribution of different subset samples. Why not analyze the distribution of the total sample? For example, in Figures 1 and 2, add a histogram of the total sample, and check for any changes in the conclusion about the distribution.
Thank you for your suggestions! We have added the total samples in Figure 1 and Figure 2. Among all the samples with real mass and radius, the planets discovered by the transit method dominate the distribution of the total samples, so the general distribution seems to have no special variation.
(3) In “Section 2.3.4 Planetary radius and period (radius valley)”, Line 260-261, the authors write "excluding stars (M<0.45 M⊕)," but in “Section 2.4 The mass of the host star and its planet”, why do the authors not exclude the low-mass M-type stars (see Figure 8)? And Figure 8 shows that there are some data points with significant errors. Please explain them.
Thank you for your suggestions! In section 2.3.4, because the formation mechanism of planets around M dwarfs may be different from that around FGK stars, which may have an impact on the occurrence rate of planets with different radius, we exclude samples with M<0.45M⊕ in section 2.3.4. In section 2.4, we will discuss the planetary occurrence rate under the stellar spectral type, so we have not excluded the samples of M dwarfs.
I have some other small suggestions.
- Figure modification:
(1) In Figure 4, the horizontal axis can be added with the error bar represented by standard deviation, to give readers credibility of the relationship trend.
Thank you for your suggestions! We added this point in Figure 4 and made a note at the label.
(2) In Figure 8, the upper right panel and right panel are plotted separately, not within the main panel.
Thank you for your suggestions! We repainted the subgraph on a separate panel.
The left panel of Figure 8 displays the relationship between the mass of planets and the mass of host stars, the right panel displays the metallicity of these host stars, and the lower panel displays the average number of planets owned by each host star, the number shown is the number of samples. The red square is type A, the blue triangle is type F, the purple star is type G, the green hexagon is type K, and the black quadrangle is type M. The gray area on the left panel is M-type stars with planets with mass larger than 100 M⊕, and their average mass and metallicity are displayed on the black dot on the right panel (the red error bar is the standard deviation). The narrow orange area in the left panel corresponds to the right panel, it shows the cutoff mass of the K-type host star when its metal abundance jumps, which is about 0.8 M☉.
- The meaning of the parameters in Table 3, such as Tsky, and Tsys.
Thank you for your suggestions! Where Tsky is the antenna temperature of the radio telescope and Tsys is the system temperature of the radio telescope. We have added these in the paper.
- Some confusing sentences:
(1)In Line 113, what does “cycle” mean?
Thank you for your suggestions! We misspelled the word, and we have removed it
(2)In Line 132, “… but due to the limited aperture of space facilities, the sensitivity to large-mass planets has declined.” Can the authors provide a detailed explanation of the reason?
Thank you for your suggestions! We deleted this unverified sentence.
(3)In Line 327-330, the authors only explained the bias of RV detection and it is need to add the bias of transition detection.
Thank you for your suggestions! For the transit method, people tend to look for planets in star systems with high metallicity and nearby bright sun-like star systems, so there are more investigations on G-type stars, which may also reduce the probability of finding planets in F-type stars. We added this to the paper.
Lines : 324-327
“For the transit method, people tend to look for planets in star systems with high metallicity and nearby bright sun-like star systems, so there are more investigations on G-type stars, which may also reduce the probability of finding planets in F-type stars.”
(4)In Figure 8, the authors mentioned “the cutoff mass of the K-type host star when its metal foundation jumps,” Please describe the causes of this phenomenon.
Thank you for your suggestions! This seems to be just an observation phenomenon, and we can't find relevant theories to explain it in detail. However, it is worth noting that the jump of the metal foundation of the K-type host star at the cut-off mass does not lead to a similar phenomenon in the mass distribution of the planets (see the orange truncation area). It is possible that K-type stars need higher metal foundation to form more massive planets.
The gray area on the left panel is M-type stars with planets with mass larger than 100M⊕, and their average mass and metallicity are displayed on the black dot on the right panel (the red error bar is the standard deviation).
(5)In Line 333-334, the authors mentioned “Small-mass and metal-rich M-type stars with high planetary ownership rates and can also have mass planets,” but I cannot obtain the information on “metal-rich” from the figures or text. Please provide a detailed explanation.
Thank you for your suggestions! In the systems with planets with mass larger than 100M⊕, the average host mass is 0.519M☉ and the average metallicity reaches 0.158 dex. We added this to the paper.
Lines:332-334
“Among the M-type host stars with planets with mass greater than 100M, the average host star mass is 0.519 M⊙, but the average metallicity reaches 0.158 dex”
(6)What is the meaning of “rows of data or exoplanets”? For example, in Line 72 and Line 426.
Thank you for your suggestions! We changed “A total of 32112 rows of data ” to “A total of 32,112 samples containing duplicates”
Lines:73
“we used the data containing duplicates from 32,112 of 4,933 exoplanets.”
Lines:428-429
“A total of 32,112 samples containing duplicates were used to investigate statistical distribution analysis of exoplanet and host star properties, as well as their relationships.”
Thank you for your hard work again. We revised all of them. It is urgent and important for Li Baoda to use the manuscript to apply for the Master degree.
Best Wishes
Zhang liyun
Guizhou University
